# Sensing of HIV-1 by TLR8 activates human T cells and reverses latency

Hany Zekaria Meås[1,2,8], Markus Haug [1,2,8], Marianne Sandvold Beckwith [1], Claire Louet [1], Liv Ryan[1], Zhenyi Hu[3,4], Johannes Landskron [4], Svein Arne Nordbø [1,5], Kjetil Taskén[4,6,7], Hang Yin [3], Jan Kristian Damås [1,2] & Trude Helen Flo [1,2,4]*

During HIV infection, cell-to-cell transmission results in endosomal uptake of the virus by target CD4+ T cells and potential exposure of the viral ssRNA genome to endosomal Toll-like receptors (TLRs). TLRs are instrumental in activating inflammatory responses in innate immune cells, but their function in adaptive immune cells is less well understood. Here we show that synthetic ligands of TLR8 boosted T cell receptor signaling, resulting in increased cytokine production and upregulation of surface activation markers. Adjuvant TLR8 stimulation, but not TLR7 or TLR9, further promoted T helper cell differentiation towards Th1 and Th17. In addition, we found that endosomal HIV induced cytokine secretion from CD4+ T cells in a TLR8-specific manner. TLR8 engagement also enhanced HIV-1 replication and potentiated the reversal of latency in patient-derived T cells. The adjuvant TLR8 activity in T cells can contribute to viral dissemination in the lymph node and low-grade inflammation in HIV patients. In addition, it can potentially be exploited for therapeutic targeting and vaccine development.

[1] Centre of Molecular Inflammation Research, Department of Clinical and Molecular Medicine, Norwegian University of Science and Technology, Trondheim, Norway. [2] Department of Infectious Diseases, St. Olavs Hospital, Trondheim, Norway. [3] School of Pharmaceutical Sciences, Tsinghua University—Peking University Joint Center for Life Sciences, Beijing Advanced Innovation Center for Structural Biology, Tsinghua University, 100082 Beijing, China. [4] Centre for Molecular Medicine Norway, Nordic EMBL Partnership, University of Oslo and Oslo University Hospital, Oslo, Norway. [5] Department of Medical Microbiology, St. Olavs Hospital, Trondheim, Norway. [6] Department of Cancer Immunology, Institute of Cancer Research, Oslo University Hospital, Oslo, Norway. [7] K.G. Jebsen Centre for Cancer Immunotherapy, Institute of Clinical Medicine, University of Oslo, Oslo, Norway. [8] These authors contributed equally: Hany Zekaria Meås, Markus Haug. *email: trude.flo@ntnu.no

The introduction of antiretroviral therapy (ART) has transformed human immunodeficiency virus type 1 (HIV-1) infection from a lethal disease to a chronic condition, but there is still no cure for HIV infection. Although ART often restores peripheral CD4+ T cell counts, persistent immune dysfunction and inflammation strongly predict risk of non-AIDS morbidity and mortality[1–3]. Enhanced inflammation is also apparent in elite controllers who show no detectable levels of HIV RNA in the absence of ART. The cause and cellular source(s) of chronic inflammation associated with HIV-1 have not been fully elucidated. Inflammatory responses are best characterized and most potently induced by innate immune cells like monocytes, macrophages, and dendritic cells (DCs). Less is known about the responses of innate immune receptors in CD4+ T cells, despite the fact that T cells also express classical pattern-recognition receptors (PRRs) like Toll-like receptors (TLRs) and inflammasomes, suggesting that they can directly recognize and respond to microbial and danger molecules[4]. HIV-1 targets immune cells and innate immune sensors are thus remarkably positioned at the nexus of viral replication and immunity.

HIV-1 infects a number of cell types. However, viral replication mainly occurs in CD4+ T cells. HIV-1 entry into target cells requires the attachment of the viral envelope glycoprotein gp120 to CD4, followed by CXCR4 or CCR5 co-receptor-mediated fusion with the host cell membrane. The choice of co-receptor is dictated by viral tropism: R5-tropic HIV dominates in the early stages of infection while X4-tropic virus emerges in advanced disease[5]. Direct cell transfer of the virus from an HIV-1-infected cell to a juxtaposed target cell has been shown to be the predominant mode of transmission. It is estimated that infection by cell-associated virus is 18,000-fold more efficient than cell-free virus when compared in vitro[6], and based on experimental and mathematical models, cell-to-cell transmission is thought to contribute to 60% of total virus infection[7]. HIV dissemination occurs in lymphoid tissues where a high concentration of T cells is present. DCs are thought to seed HIV infection in the lymph nodes by transmitting the virus to T cells during antigen presentation[8]. However, productively infected T cells are highly migratory and can disseminate the infection through cell-to-cell transmission locally in the lymph node. In addition, recirculation of HIV-infected T cells is important for the establishment of systemic infection[9]. Physical contact between donor and target cell results in the formation of a virological synapse (VS), stabilized by the interactions of HIV-Env/CD4 and adhesion molecules, which triggers uptake of virus by the target cell[10]. Endosomal uptake of HIV-1 relies on the binding of HIV-gp120 to CD4 and is independent of engagement of CXCR4 or CCR5 co-receptors[11]. However, in the presence of the appropriate co-receptor, the virus can enter the cytoplasm by fusing with endosomal membranes resulting in productive infection of the target cells[12]. Failure to engage the co-receptor due to inappropriate virus tropism can lead to recycling of the virus to the cell surface[11,13] as well as inactivation and degradation in endosomal compartments[14–16]. It is currently not clear to what extent HIV-1 activates CD4+ T cell PRRs during the various stages of infection, and how this would impact infection. In this situation, the local cytokine environment may well be shaped by T cells and act in an autocrine/paracrine manner.

Ten functional TLRs have been discovered in humans of which TLR3, TLR7, TLR8, and TLR9 are located in endosomes and lysosomes[17]. TLR7 and TLR8 recognize single-stranded RNA (ssRNA) in the forms of degradation products, nucleosides, and oligoribonucleotides[18–22] while TLR3 and TLR9 bind double-stranded RNA (dsRNA) and double-stranded DNA (dsDNA), respectively[23,24]. Activated TLR dimers recruit adaptor proteins MyD88 (myeloid differentiation primary response 88) or TRIF (Toll/interleukin-1 (IL-1) receptor domain-containing adapter protein inducing interferon-β (IFN-β)) which signal through sequential activation of IRAK (IL-1 receptor-associated kinase) family kinases, IKKs (inhibitor of κB kinase), and MAPKs (mitogen-activated protein kinase), leading to the activation of the transcription factors NF-κB and AP-1-driving inflammatory cytokines, or interferon regulatory factors (IRFs) driving type I IFNs[25]. Several disease-causing viruses interact with endosomal TLRs including West Nile virus, Epstein-Barr-virus, influenza and HIV-1[26]. HIV-1 is an enveloped retrovirus with two copies of a ssRNA genome that may be recognized by TLR7 and TLR8[27–29]. HIV endocytosis by plasmacytoid DCs induces TLR7 activation and type I IFN production[30]. In monocytes and myeloid DCs, TLR8 has been shown to recognize HIV ssRNA resulting in MyD88-dependent activation and subsequent interleukin (IL)-1β production[31–35]. Several studies show expression of TLR7 and TLR9 in CD4+ T cells[36–39] whereas inconsistent results are reported for TLR8: some studies claim that TLR8 is present in T cells[38,40], while others show the absence of the receptor[39,41]. In a study by Dominguez-Villar et al.[42], TLR7 stimulation increased HIV replication in productively infected CD4+ T cells while inhibition of TLR7 had the opposite effect, suggesting a role for TLR7 in the regulation of the viral life cycle[42]. Here, we reveal that HIV-1 is endocytosed and recognized by TLR8 in human primary CD4+ T cells and that TLR8 stimulation induces an inflammatory response that favors HIV-1 replication and reversal of latency. Our results uncover a function for TLR8 in human primary CD4+ T cells as adjuvant in activation of inflammatory cytokines and in HIV replication.

## Results

**HIV is endocytosed in human primary CD4+ T cells**. We first set out to generate a cell-to-cell transmission model to induce HIV-1 endocytosis in human CD4+ T cells and to follow intracellular trafficking of the virus when viral membrane fusion is blocked, preventing the establishment of productive infection. Most peripheral CD4+ T cells express CXCR4 (Supplementary Fig. 1d) but both X4- and R5-tropic virus were utilized. Primary CD4+ T cells from HIV-negative donors were activated by TCR stimulation using anti-CD3 coated plates and soluble anti-CD28 for 48 h. Activated CD4+ T cells were co-cultured for 3 h with donor HeLa or HEK293T cells expressing the X4 HIV-1-Gag-iGFP, human monocyte-derived macrophages (MDMs) infected with R5 HIV-1-Gag-iGFP or donor CD4+ T cells infected with X4-Gag-iGFP in the presence of the CXCR4 inhibitor AMD3100 or the CCR5 inhibitor Maraviroc, to prevent co-receptor mediated HIV-1 fusion. T cells were trypsinized prior to analysis to remove extracellular membrane-bound virus. In agreement with previous studies[11,43], confocal microscopy revealed that HIV-1 accumulated in VS and was transferred to intracellular (trypsin-resistant) compartments in the target CD4+ T cell (Fig. 1a, b). Transfer of virus also occurred at VS if primary HIV-infected MDMs or CD4+ T cells were used as donor cells (Fig. 1a). Quantification revealed an average of close to 30 intracellular virus puncta per acceptor T cell after 24 h of co-culture (Fig. 1b). The trypsin-resistant HIV-1 compartments only occasionally stained positive for markers of early endosomes (EEA1) or late endosomes/lysosomes (LAMP1) (Supplementary Fig. 1b). We further utilized flow cytometry to quantify the frequency of CD4+ T cells harboring intracellular HIV-1 (Fig. 1c and Supplementary Fig. 1c), which decreased over time when fusion was inhibited by AMD3100 (Fig. 1c). Inhibition of uptake by pretreatment with the dynamin inhibitor Dynasore markedly reduced the amount of CD4+ T cells harboring trypsin-resistant HIV (Fig. 1d). In addition, we compared the efficiency of

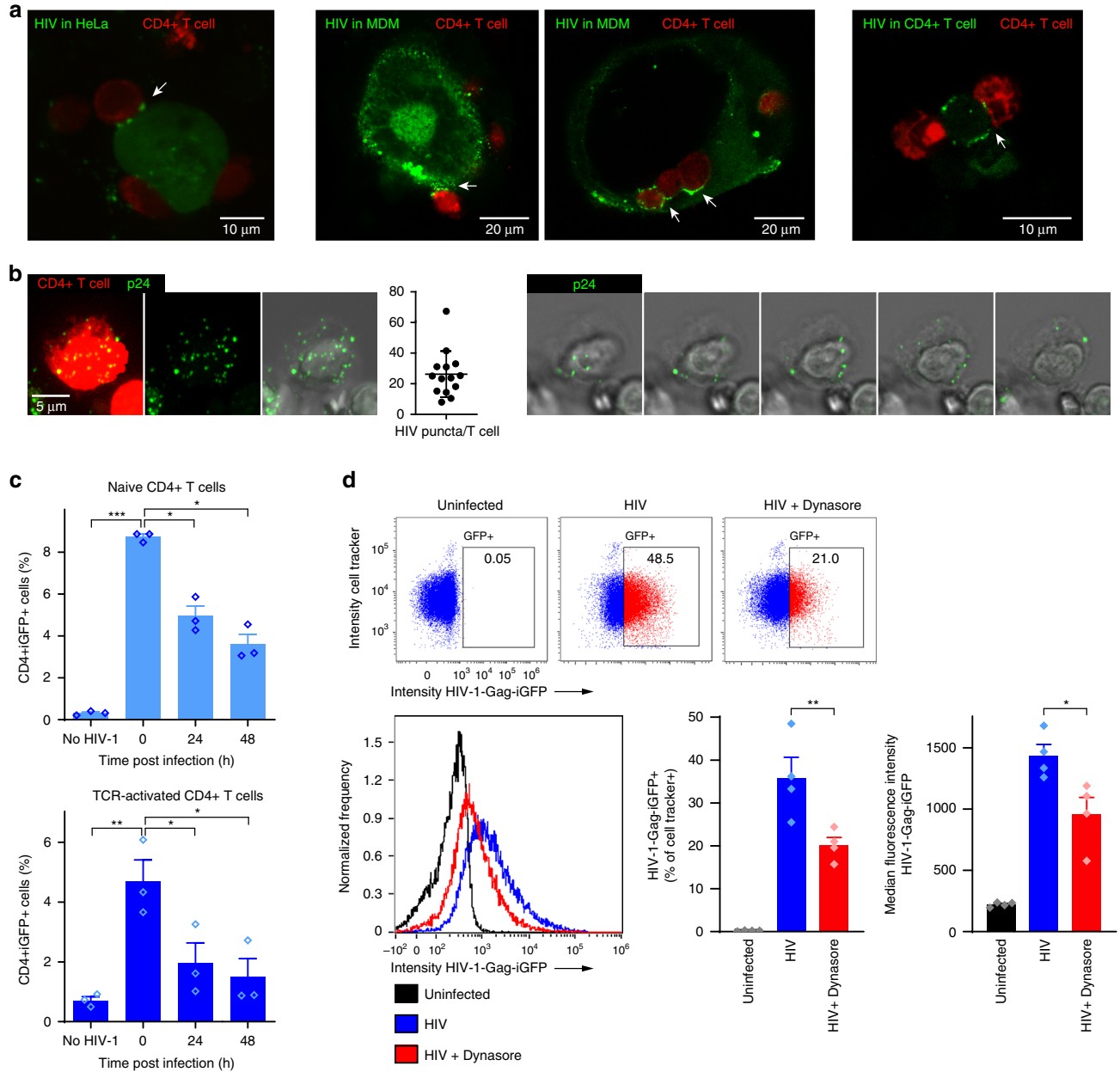

**Fig. 1 HIV-1 is endocytosed and degraded by CD4+ T cells. a** HeLa cells expressing X4-tropic HIV-1-Gag-iGFP (green), primary human monocyte-derived macrophages (MDM) infected with cell-free HIV-1-Gag-iGFP_JRFL (R5 HIV-1), or primary CD4+ T cells infected with cell-free HIV-1-Gag-iGFP (X4 HIV-1) were co-cultured for 3 h with TCR-activated CD4+ T cells (Cell Tracker Deep Red) in the presence of the CXCR4 inhibitor, AMD3100 or the CCR5 inhibitor, Maraviroc. HIV-1 is accumulating in the virological synapse between donor and acceptor cell for all tested donor cell types. **b** CD4+ T cells were co-cultured with HEK293T cells expressing NLENG1-IRES (X4 HIV-1) for 24 h, trypsin treated for removal of extracellular virus and labeled with antibodies against HIV-1 p24. Maximum intensity projection images of a representative CD4+ T cell (stained with Cell Tracker Deep Red, left image panel), and quantification of the number of trypsin-resistant HIV puncta (identified by p24 staining, green) per acceptor T cell. Slices from a confocal z-stack through the same cell depicting HIV puncta in trypsin-resistant compartments (right image panel, every second slice is shown). **c** Frequency of trypsinized HIV-1-positive CD4+ T cells after co-cultured for 3 h with HIV-1-Gag-iGFP-expressing HeLa cells, analyzed by flow cytometry 0, 24, and 48 h post co-culture. Bars represent mean + SEM from three independent experiments. **d** CD4+ T cells were pre-treated with 80 μM Dynasore for 1 h prior to co-culture with HEK293T cells expressing HIV-1-Gag-iGFP (X4 HIV-1) for 24 h and analysis by imaging flow cytometry. Dot plot examples showing frequencies of trypsinized HIV-1-positive CD4+ T cells (top). Histogram overlay of median fluorescent intensity of Gag-GFP (left) and quantification (right). Bars represent mean + SEM from four independent experiments. Statistical significance was determined from log-transformed data by repeated measures two-way ANOVA with Dunnett's post-test in **c** and two-tailed paired *t*-test in **d**; significance levels: *p < 0.05; **p < 0.01; ***p < 0.001. Source data are provided as a Source Data File.

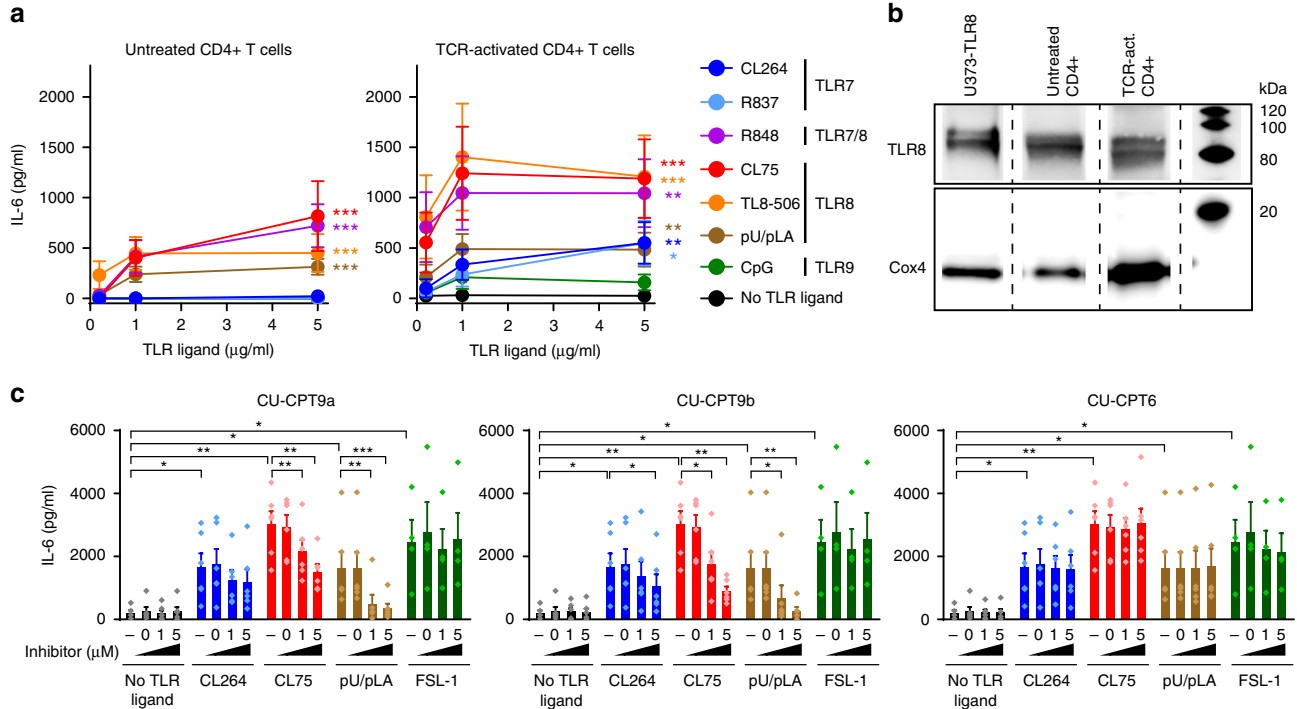

**Fig. 2 TLR8 is expressed and induces secretion of IL-6 in human primary CD4+ T cells. a** Untreated or TCR-activated human primary CD4+ T cells were stimulated with 0.2, 1, and 5 µg/ml ligands to TLR7 (CL264, R837), TLR7/8 (R848), TLR8 (CL75, TL8-506, pU/pLA), or TLR9 (CpG, 0.2, 1, 5 µM) for 24 h and IL-6 secretion was analyzed by ELISA. Data represent mean + SEM from four independent experiments. **b** TLR8 expression in untreated or TCR-activated CD4+ T cells was analyzed by immunoblotting. Lysates from U373 cells overexpressing human TLR8 were used as positive control and Cox4 as housekeeping control. **c** CD4+ T cells were untreated (−) or pre-treated with DMSO (=0), 1 or 5 µM of TLR8-specific inhibitors CU-CPT9a, CU-CPT9b, or the negative control compound CU-CPT6, for 2 h prior to TCR activation and stimulation with 5 µg/ml CL264, CL75, pU/pLA, or FSL-1 (TLR2) for 24 h. IL-6 secretion was analyzed by ELISA. Bars represent mean + SEM from six independent experiments. Statistical significance was determined in **a** by calculating areas under the curve (AUC) followed by repeated measures one-way ANOVA with Dunnett´s post-test on log-transformed data, in **c** by two-way ANOVA with Dunnett's post-test; significance levels: *$p < 0.05$; **$p < 0.01$; ***$p < 0.001$. Source data are provided as a Source Data File.

cell-associated and cell-free virus in infecting primary CD4+ T cells. SupT1 cells were infected with eGPF+ HIV-1 prior to co-culture with TCR-activated CD4+ T cells in the presence or absence of a 0.4 µm trans-well membrane barrier allowing only cell-free virus to pass. In agreement with previous studies, infection with cell-associated virus resulted in significantly higher infection rate of CD4+ T cells when compared with cell-free virus (Supplementary Fig. 1a). Our results thus indicate that HIV-1 is efficiently endocytosed upon cell-to-cell transfer and, in the absence of co-receptor binding, later degraded within the acceptor T cell.

**CD4+ T cells secrete IL-6 in response to TLR8 ligands**. The single-stranded RNA genome of HIV represents a potential ligand for the RNA-sensing endosomal TLR7 and TLR8. However, endosomal TLR responses in T cells are poorly characterized so we initially assessed functional endosomal TLR responses in primary CD4+ T cells using synthetic ligands. CD4+ T cells were purified from HIV-negative donors, yielding 95.7% CD3+CD4+ T cells with less than 1% monocytes or DCs (Supplementary Fig. 2a, b). Untreated or TCR-activated CD4+ T cells were challenged with increasing concentrations of TLR7 ligand CL264, TLR7/8 ligand R848, TLR8 ligands CL75, polyU/poly ʟ-arginine (pU/pLA) or TL8-506, or TLR9 ligand CpG for 24 h and assessed for IL-6 production by ELISA (Fig. 2a). IL-6 was secreted by resting CD4+ T cells in response to all TLR8 ligands while TLR7 or TLR9 ligands did not induce IL-6 secretion (Fig. 2a, left). In TCR-activated T cells, concomitant treatment with TLR8 ligands

further increased IL-6 secretion (Fig. 2a, right): at low doses (0.2–1.0 µg/ml), TLR8 and TLR7/8 ligands induced 7 to 30 times higher IL-6 levels relative to the TLR7 ligands R837 and CL264 (Fig. 2a, right). TLR7 ligands did increase IL-6 secretion at high doses (5 µg/ml), but this was not the case for the TLR9 ligand CpG. TLR8 ligands also induced IL-6 production from ultrapure T cells (>99.5% CD4+ T cells; Supplementary Fig. 2c). Addition of 0.3% CD14+ monocytes (corresponds to the maximum of CD14+ cells that we see in isolations) or (CD4+ T cell-depleted) peripheral blood mononuclear cells (PBMCs) did not significantly affect TLR8-induced IL-6 or IFNγ production from CD4+ T cells (Supplementary Fig. 2d, e). Higher amounts of contaminating cells affected IL-6 but not IFNγ production (Supplementary Fig. 2d, e). Thus, IL-6 and T cell intrinsic IFNγ was mainly contributed from CD4+ T cells.

Our findings suggest that CD4+ T cells respond to TLR8 ligands with inflammatory cytokine production, less so to TLR7 ligands and not to TLR9 ligands. This could be due to differential receptor expression. We made use of a recently published data set generated from high-resolution mass-spectrometry-based proteomics of FACS-sorted immune cells to assess the expression of TLR7, TLR8, TLR9, and MyD88 in resting and TCR-activated CD4+ T cells[44]. The majority of CD4+ T cell subsets were found to express these proteins, albeit at variable levels (Supplementary Fig. 3). We thus purified primary human CD4+ T cells and used immunoblotting to determine TLR8 protein expression. We were able to detect two bands corresponding to cleaved TLR8 in untreated and TCR-activated CD4+ T cells (Fig. 2b).

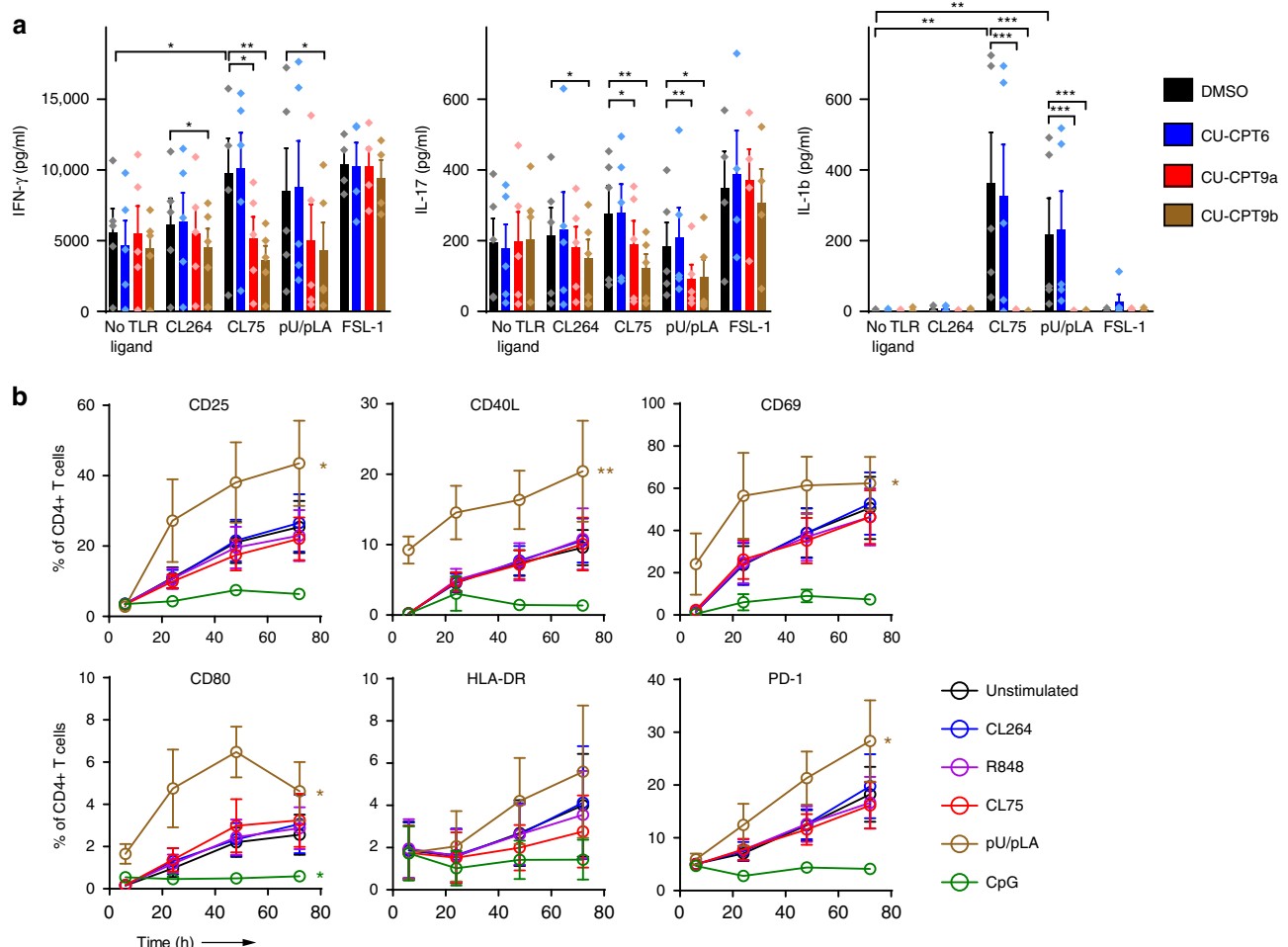

**Fig. 3 TLR8 enhances cytokine secretion and surface activation markers in CD4+ T cells. a** CD4+ T cells were pre-treated with 5 µM TLR8-specific inhibitors CU-CPT9a, CU-CPT9b, or the negative control compound CU-CPT6, prior to TCR activation and stimulation with synthetic TLR ligands (5 µg/ml). Cell supernatants were harvested at 24 h and IFN-γ, IL-17, and IL-1β analyzed using multiplex ELISA. Bars represent mean values + SEM from five independent experiments. **b** TCR-activated CD4+ T cells were stimulated with 5 µg/ml CL264, R848 (TLR7/8), CL75, pU/pLA, or CpG (TLR9, 5 µM). T cell activation markers were analyzed by flow cytometry at the indicated time points after surface staining for CD25, CD40L, CD69, CD80, HLA-DR, and PD-1. Graphs represent mean values ± SEM from three independent experiments. Statistical significance was determined in **a** by ANOVA with Dunnett's post-test, in **b** by area under the curve (AUC) calculation followed by one-way ANOVA with Dunnett's post-test on log-transformed data; significance levels: *$p < 0.05$; **$p < 0.01$; ***$p < 0.001$. Source data are provided as a Source Data File.

To further assess the specificity of the TLR8 response, we utilized TLR8 inhibitors that stabilize the TLR8 dimer in its inactive state[45,46]. CD4+ T cells were pre-treated with the structurally optimized TLR8 inhibitors CU-CPT9a and CU-CPT9b, or the negative control compound CU-CPT6, for 2 h prior to TCR activation and stimulation with CL264, CL75, pU/pLA or the TLR2 ligand FSL-1 for 24 h. TLR8-induced IL-6 was significantly reduced by inhibitors CU-CPT9a and CU-CPT9b (Fig. 2c). Inhibitor CU-CPT9b was superior compared to CU-CPT9a and almost completely blocked IL-6 secretion at the highest concentration used (5 µM). As expected, the TLR8 inhibitors had no effect on cells treated with the TLR2 ligand FSL-1, whereas some reduction of CL264 induced IL-6 was observed at high concentrations of CU-CPT9b (Fig. 2c). Taken together our data confirm that human primary CD4+ T cells express TLR8 and respond to TLR8 ligands by secreting IL-6.

**TLR8 stimulation potentiates TCR-activated CD4+ T cells.** To more broadly assess the impact of endosomal TLRs on CD4+ T cell responses, we characterized secretion of additional cytokines

and upregulation of T cell activation markers 24–72 h post TCR+TLR activation. Despite donor variations, we found that TLR8 ligands, but not TLR7 or TLR9 ligands, significantly enhanced the secretion of a wide range of cytokines over time (IFN-γ, GM-CSF, TNF-β, IL-10, IL-12, IL-17, IL-1β) in addition to IL-6 (Supplementary Fig. 4). In fact, CpG had the opposite effect and acted antagonistic rather than augmenting TCR activation. Select cytokine responses were investigated further using the TLR8 inhibitors. Pre-treating the cells with CU-CPT9a and CU-CPT9b completely inhibited the secretion of IFN-γ, IL-17 and IL-1β induced by CL75 and pU/pLA, but not the TLR2 ligand FSL-1, thus confirming TLR8 specificity (Fig. 3a).

Engagement of the TCR results in upregulation of surface activation molecules that play an important role in T cell effector functions. Only one of the TLR8 ligands, pU/pLA, significantly increased the frequencies of CD4+ T cells expressing CD25, CD40L, CD69, CD80, and programmed cell death protein 1 (PD-1) compared to TCR activation alone (Fig. 3b). CL75, CL264, and R848 showed no effect, suggesting differential responses even to different TLR8 ligands. Of note, the TLR9 ligand CpG prevented TCR-mediated upregulation of activation markers,

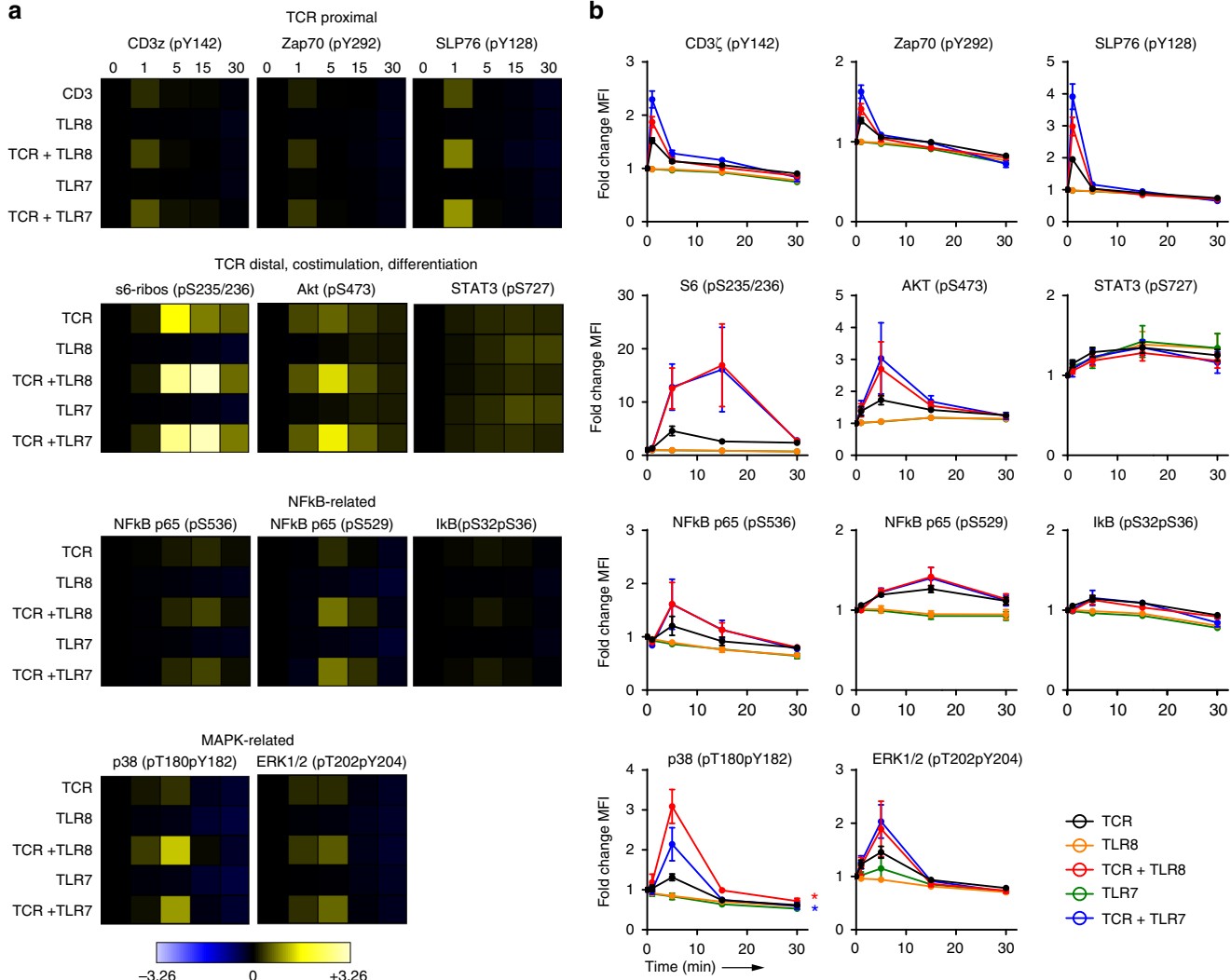

**Fig. 4 TLR8 and TLR7 augment TCR signaling in CD4+ T cells.** TCR signaling was induced in CD4+ T cells for 0–30 min in the presence or absence of 5 μg/ml of CL264 (TLR7), CL75 (TLR8), or both. Phosphorylation of signaling proteins in the TCR and TLR pathways was analyzed using phospho-epitope-specific antibodies. TCR proximal (CD3ζ, Zap70, SLP76) and more distal (ribosomal protein S6, Akt, STAT3) signaling proteins were included, as well as proteins in the NF-κB (p65, IκB) and MAPK (p38, ERK1/2) pathways. Phosphorylation levels were quantified as arcsinh ratios of the median fluorescence intensity for the various phospho-specific Abs in activated versus unstimulated CD4+ T cells. **a** Phosphorylation levels displayed as heat maps (blue = reduced, yellow = increased) for one representative out of three experiments. **b** Phosphorylation levels from all three experiments displayed as the average arcsinh ratio± SEM of median fluorescence intensities. Statistical significance was determined by area under the curve (AUC) calculation followed by one-way ANOVA with Dunnett's post-test on log-transformed data; significance levels: *$p < 0.05$; **$p < 0.01$; ***$p < 0.001$. Source data are provided as a Source Data File.

which is in line with the inhibitory effect seen on cytokine secretion (Supplementary Fig. 4). However, it was only significant for CD80 expression. Taken together, our results show that engagement of TLR8, but not TLR7 or TLR9, broadly enhances CD4+ T cell activation phenotype and cytokine secretion.

**TLR8 and TLR7 augment TCR mediated signaling in CD4+ T cells.** Signaling from endosomal TLR7/8/9 is best characterized in antigen-presenting cells (APCs) such as macrophages and DCs[25]. Receptor activation culminates in nuclear translocation of transcription factors NF-κB, MAPK/AP-1, and IRFs with subsequent production of inflammatory mediators and/or type I IFNs. NF-κB and MAPK signaling pathways are also involved in TCR signaling and we next addressed if TCR and TLR signaling pathways in CD4+ T cells interfere with each other. CD4+ T cells were activated (cross-linking of anti-CD3/CD28 with avidin) in the presence or absence of TLR7 and TLR8 ligands for

0–30 min. Phosphorylation of key signaling proteins in the TCR and TLR signaling pathways was assessed by phospho-flow cytometry (Fig. 4)[47]. TCR proximal signaling molecules CD3ζ, Zap70, and the adaptor protein SLP76 was transiently phosphorylated 1 min post TCR stimulation and only moderately increased by TLR7 or TLR8 ligands. A less studied TCR signaling branch involves ribosomal protein S6 (S6), which is assumed to play a role in cell cycle progression in T cells[48]. S6 phosphorylation peaked between 5 and 15 min post TCR stimulation, and concomitant engagement of TLR7/8 increased signaling (2.7–6.4-fold compared to TCR stimulation alone), though kinetics did not change significantly.

Further downstream, the TCR signal branches into different pathways of which the Ca$^{2+}$/nuclear factor of activated T cells (NFAT), NF-κB, and MAPK pathways are of central importance. This is also where we would expect most cross-talk from concomitant TLR engagement. Phosphorylation of the more

distal proteins in the MAPK (MAPK p38 and ERK1/2) and NF-κB (NF-κB p65) pathways was highest 5 min post TCR activation. Additional TLR7/8 stimulation significantly increased phosphorylation of p38 (2.3-fold) but not ERK1/2 and NF-κB pS536, although the tendency was the same (1.3-fold for ERK1/2 and NF-κB pS536). No activation of NF-κB p65 phosphorylation was observed by engagement of TLR7 or TLR8 alone. Taken together, the main immediate effect of TLR7/8 engagement seemed to be adjunctive in strengthening some of the TCR signaling pathways, such as ribosomal protein S6 and pathways that cross with canonical TLR signaling like the MAPK and NF-κB pathways.

**TLR8 promotes differentiation to Th1 and Th17 cells.** Activated CD4+ T cells proliferate and eventually differentiate into different lineages of T helper (Th) cells depending on the cytokine environment. APCs are a major source of lineage-polarizing cytokines, but our results suggest that T cells themselves could be contributing since IL-6 is known to induce expression of RORγT (retinoic acid-related orphan receptor γT) which drives Th17 differentiation. To test this hypothesis, we compared the production of signature cytokines associated with different effector T cell lineages in TCR + TLR-activated CD4+ T cells over time using intracellular flow cytometry (Fig. 5a–c). TLR8 stimulation significantly increased production of Th1 cytokine IFN-γ and Th17 cytokine IL-17 compared to TCR activation alone, whereas no effect was seen on the Th2 cytokine IL-4, or on TNF-α and IL-2 (Fig. 5c). TLR7 or TLR9 ligands did not affect cytokine production. To further verify that the observed responses were from TLR ligands acting directly on T cells, we added up to 3% CD14+ monocytes or PBMCs depleted of CD4+ cells to the purified CD4+ T cells before TCR+TLR stimulation (Supplementary Fig. 2e). The addition of monocytes or CD4+-depleted PBMCs did not influence on TLR8-induced IFN-γ production from CD4+ T cells. However, under these conditions the T cells responded to CpG with IFN-γ production, most likely indirectly via CpG activation of monocytes or DCs. Finally, in agreement with other studies[39], we found that CD4+ T cells with a memory phenotype (CD45RO+CD45RA−) responded with higher IFN-γ-production in response to TCR+TLR8 stimulation than naïve cells (CD45RO−CD45RA+) (Supplementary Fig. 5a).

Pre-treatment of TCR-activated CD4+ T cells with TLR8 inhibitors efficiently decreased IFN-γ and IL-17 induced by TLR8 ligands but did not affect TLR2 (FSL-1)-mediated effector cytokine responses (Fig. 5d). TLR7 ligand CL264 did not induce IFN-γ and IL-17 producing T cells, and TNF-α levels were unaffected by TLR7/8 engagement or the inhibitors (Fig. 5c, d). The adjuvant effect of TLR8 signaling on differentiation of CD4+ T cells towards the Th1/Th17 axis was confirmed in two additional sets of experiments: Expression of lineage-specific transcription factors (T-bet and RORγt, Supplementary Fig. 5b) and effector cytokine production in CD4+ T cells re-stimulated 8 days after TCR+TLR activation (Supplementary Fig. 5c). Th1/Th17 cells are central in defense of intracellular infections, and proposed to support HIV-1 long-term persistence in patients receiving ART[49].

**Endosomal HIV induces inflammatory cytokines in CD4+ T cells.** Endocytosed HIV-1 can potentially activate TLRs in the endosome. To examine if endosomal HIV-1 is sensed by TLR8, we co-cultured CD4+ T cells with HEK293T cells expressing X4- or R5-tropic HIV-Gag-iGFP in the presence of CXCR4 antagonist, AMD3100, or CCR5 antagonist, Maraviroc, to block fusion. CD3/CD28 activation beads were added to the co-culture and cytokines analyzed by multiplex ELISA after 24 h. Endosomal X4 HIV-1 and R5 HIV-1 both significantly increased the levels of IL-

6 and IL-17 compared to TCR activation alone, and the TLR8 inhibitors efficiently reduced (CU-CPT9a) or completely inhibited (CU-CPT9b) the responses (Fig. 6, absolute concentrations in Supplementary Fig. 6). IFN-γ responses were only modestly increased by HIV-1 and, accordingly, not significantly reduced by the TLR8 inhibitors. Our data show that endosomal HIV-1 increases cytokine production in CD4+ T cells by engaging TLR8.

**TLR8 re-activates latent HIV and enhances replication.** HIV-1 can remain latent in resting memory CD4+ T cells and it has been shown that pro-inflammatory stimuli can cause reactivation of viral replication[50–53]. Crucial factors responsible for initiating transcription at the HIV long terminal repeat are NF-κB, NFAT and AP-1, and the protein kinase C (PKC) pathway[54]. To determine whether TLR8 stimulation can enhance viral reactivation in latently infected cells, CD4+ T cells were isolated from the blood of nine HIV patients on ART with plasma HIV-1 viremia below the limit of detection (Supplementary Table 1). Activated CD4+ T cells (CD69+, CD25+, HLA-DR+) were depleted, and a viral outgrowth assay was performed by culturing resting CD4+ T cells in the presence or absence of gamma-irradiated PBMCs and the mitogenic lectin phytohemagglutinin (PHA), TLR8 ligands CL75 or pU/pLA, or the TLR7 ligand CL264, the histone deacetylase inhibitor (HDACi) SAHA, the protein kinase C (PKC) agonist Bryostatin or the PKC agonist/calcium ionophore PMA/ionomycin (Fig. 7a). Reactivation of latent virus in response to pU/pLA treatment was observed in samples obtained from all patients at days 7 and 14 post-treatment while CL75 reversed latency in cells obtained from two patients in the presence of PHA/ γ-irradiated PBMC and from one patient when acting alone. Bryostatin and PMA/ionomycin showed moderate effects. No increase in viral outgrowth was seen in response to CL264 or SAHA.

Productive HIV infection is directly related to the activation state of CD4+ T cells. Quiescent CD4+ T cells cannot support viral replication and require activation for successful completion of the viral life cycle[55–58]. We next assessed if adjuvant TLR7/8/9 stimulation impacted on HIV-1 replication during productive infection (no fusion block). CD4+ T cells were TCR-activated and treated with TLR7/8/9 ligands for 24 h before infection with NLENG1-IRES-eGFP virus. HIV-1 replication was assessed by monitoring the frequency of eGPF+CD4+ T cells by flow cytometry (Fig. 7b, c). The TLR8 ligand CL75 significantly increased the frequency of CD4+ cells expressing eGFP at day 3 and day 5, whereas TLR7 and TLR9 ligands showed no significant effect on HIV-1 replication. Taken together, these findings show that TLR8 stimulation enhances HIV-1 viral replication and reversal of latency in TCR-activated human primary CD4+ T cells.

## Discussion
HIV-1 entry can occur from endosomal compartments in addition to the plasma membrane, but if the target cell expression of CXCR4 or CCR5 co-receptors does not match the tropism of the virus, HIV-1 is trapped in endosomal compartments. Endosomal degradation of HIV-1 has been reported in human macrophages[59] but what occurs in CD4+ T cells is not clear. Using a cell-to-cell transmission model, we show that X4- and R5-tropic HIV-1 is endocytosed in human CD4+ T cells and induce TLR8-dependent secretion of inflammatory cytokines, mirroring the response obtained with synthetic TLR8 ligands. Stimulation of TCR-activated CD4+ T cells with TLR8 ligands induced a "hyper-activation" state, marked by upregulation of activation markers and increased secretion of inflammatory cytokines, and

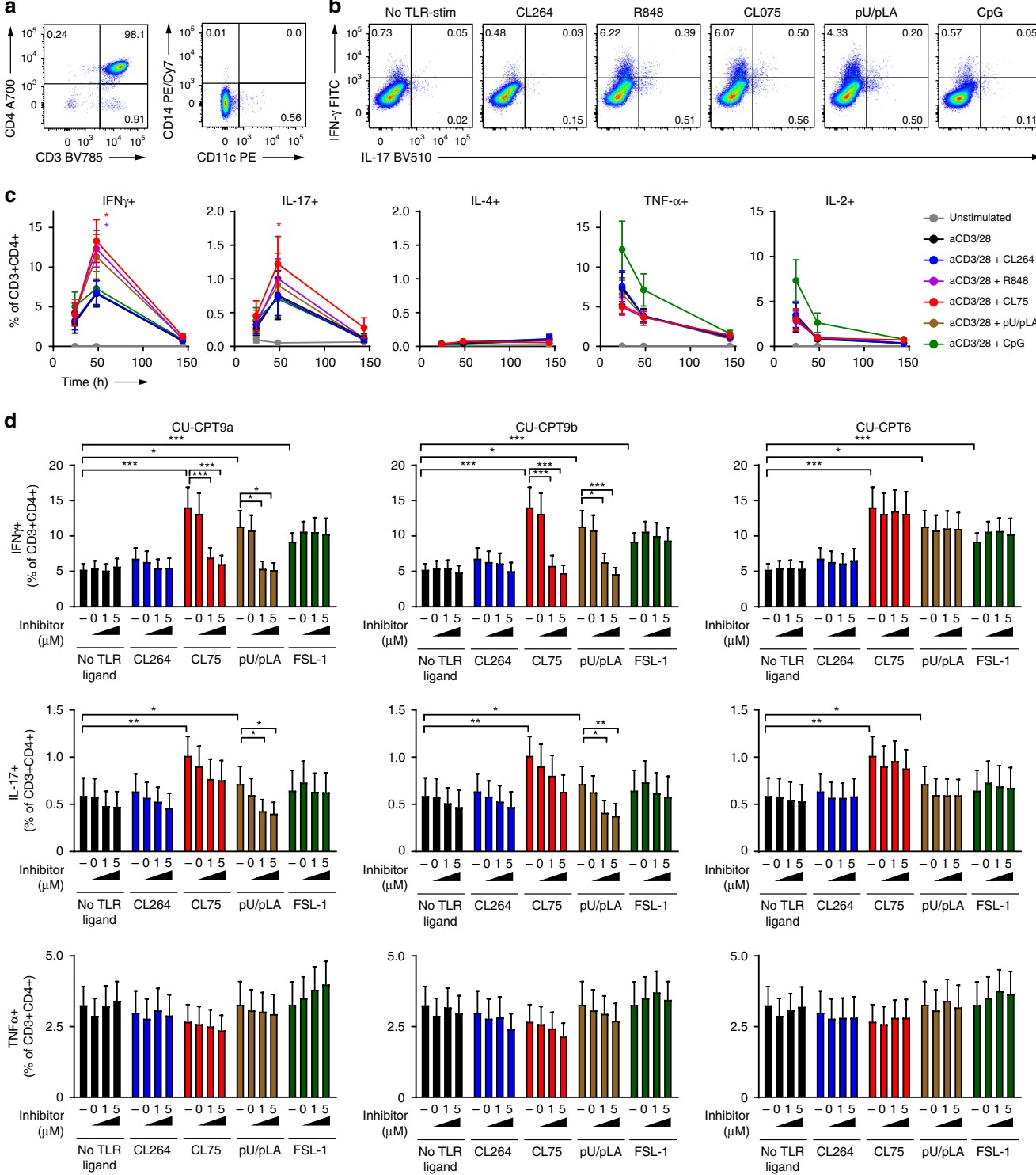

**Fig. 5 TLR8 enhances differentiation of CD4+ T cells towards Th1 and Th17. a** Purity analysis of CD4+ T cells after isolation and content of contaminating CD14+ and CD11c+ cells. **b, c** CD4+ T cells were TCR-activated in the presence or absence of 5 μg/ml CL264 (TLR7), R848 (TLR7/8), CL75, pU/pLA (TLR8), or CpG (TLR9, 5 μM). Production of CD4+ T helper effector cytokines IFN-γ, IL-17, IL-4, TNF-α, and IL-2 was analyzed at 24–144 h by intracellular flow cytometry. **b** Example of flow cytometric analysis of Th1 lineage cytokine IFN-γ and Th17 cytokine IL-17 at 48 h. **c** Frequencies of cytokine-producing CD4+ T cells over time. Results represent mean + SEM from eight independent experiments. **d** CD4+ T cells were pre-treated with TLR8-specific inhibitors CU-CPT9a, CU-CPT9b, or the negative control compound CU-CPT6 for 2 h prior to TCR activation and stimulation with 5 μg/ml of CL264, CL75, pU/pLA, or FSL-1 (TLR2) for 72 h. Cytokine production was analyzed by intracellular flow cytometry. Results from inhibitor-treated CD4+ T cells were compared to untreated (−) or DMSO (=0) treated control cells (same data in all three graphs for each cytokine). Bars represent mean + SEM from ten experiments. Statistical significance was determined from log-transformed data by repeated measures two-way ANOVA with Dunnett's post-test in **c** (48 h timepoint) and in **d**; significance levels: *$p < 0.05$; **$p < 0.01$; ***$p < 0.001$. Source data are provided as a Source Data File.

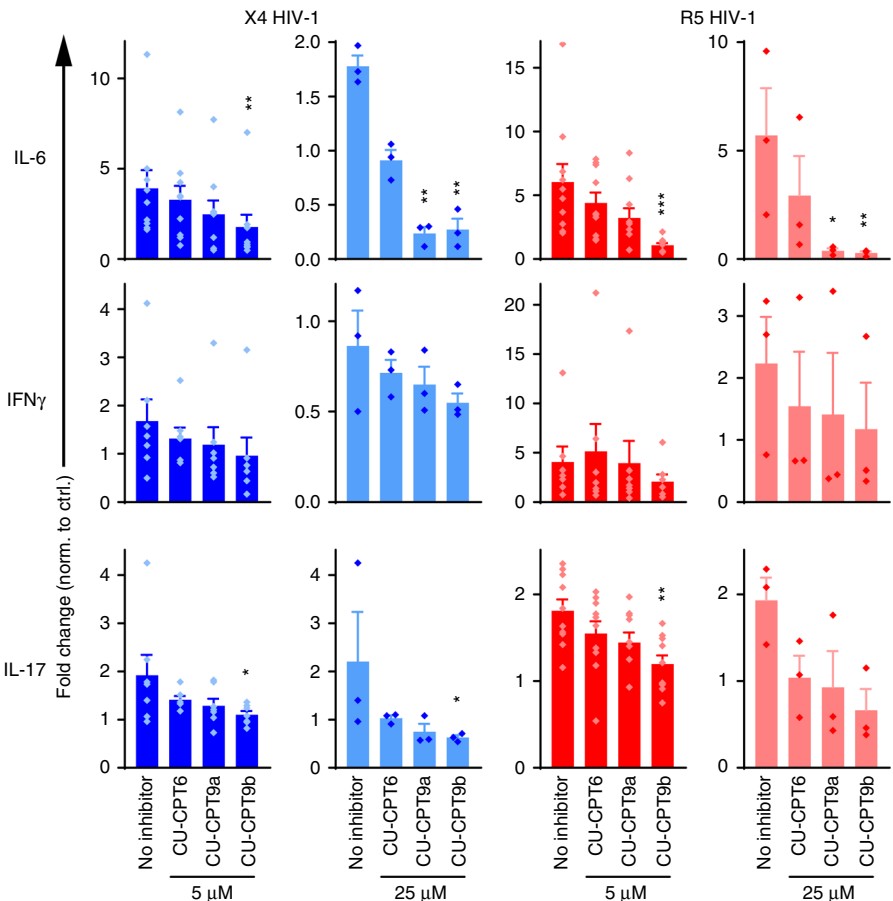

**Fig. 6 Endolysosomal HIV-1 potentiates cytokine secretion from CD4+ T cells.** CD4+ T cells were pre-treated with CXCR4 inhibitor AMD3100 or CCR5 inhibitor Maraviroc for 24 h, and TLR8-specific inhibitors CU-CPT9a, CU-CPT9b, or the negative control compound CU-CPT6 for 2 h, prior to concomitant TCR activation and co-culture with HEK293T cells expressing HIV-1-Gag-iGFP (X4 HIV-1) or HIV-Gag-iGFP JRFL (R5 HIV-1). Cytokines were analyzed in the supernatant 24 h post infection by multiplex ELISA. Results are presented as fold change relative to co-cultures in the absence of HIV-1. Bars represent means + SEM from 7 to 10 independent experiments with 5 µM inhibitor and three independent experiments with 25 µM inhibitor. Statistical significance (inhibitor samples versus untreated) was determined using repeated measures one-way ANOVA with Dunnett's post-test on log-transformed data; significance levels: *$p < 0.05$; **$p < 0.01$; ***$p < 0.001$. Source data are provided as a Source Data File.

differentiation towards pro-inflammatory Th1/Th17 effector cells. The TLR8 inhibitors CU-CPT9a and CU-CPT9b abolished the responses to TLR8 ligands demonstrating that the observed adjuvant effect was specific to TLR8. TLR8 activation also enhanced HIV-1 replication and induced reactivation of latent virus in T cells from virologically suppressed HIV patients on ART. These findings indicate a role for TLR8 in HIV-1 infection, where a "dead-end" pathway of HIV-1 endocytosis could contribute to the spread of virus in lymphoid tissues by promoting a higher degree of CD4+ T cell activation, thus priming bystander and latently infected CD4+ T cells for viral replication.

HIV-1 dissemination in CD4+ T cells is thought to be partly dependent on cell-to-cell transmission of the virus upon migration of HIV-1-laden APCs to the lymph nodes[8]. Cell-to-cell transmission occurs between a productively HIV-1-infected cell and a recipient CD4+ T cell. However, precisely assessing the contribution of cell-associated and cell-free virus to HIV disease in vivo is challenging since both modes of transmission occur concurrently and interdependently. Productively infected T cells are highly migratory and have been shown to disseminate HIV through cell-to-cell transmission locally in the lymph node of humanized mice[9]. Furthermore, recirculation of HIV-infected T cells was important for the establishment of systemic infection. In addition, simian immunodeficiency virus (SIV)-infected PBMCs isolated from rhesus monkeys penetrated the colon

epithelial layer and established new infection of host cells, while cell-free virus failed to do so[60]. Similar results were obtained in HIV infection of human colonic tissue explants[60]. Taken together, these studies suggest that cell-to-cell transmission of HIV may play a role in the dissemination of virus in vivo.

Virus tropism as well as the presence of the appropriate co-receptor on the target cells determines the fate of the virus since the co-receptor is essential for the fusion of the virion with host cell membranes. The expression pattern of CCR5 and CXCR4 on CD4+ T cells can differ in various subpopulations and over the course of infection[61,62]. Therefore, a mismatch between virus tropism and the co-receptor is plausible. Failure of the virus to enter the cytosol can lead to recycling to the cell surface or degradation[11,63]. Our data show that in the presence of fusion inhibitor, HIV is internalized into a trypsin-resistant compartment that only occasionally stains positive for early (EEA1) or late (LAMP1) endosomal markers. We observed patches of co-localized P24/EEA1 that resemble the findings by Bosch et al.[65], but overall co-localization events were few as also shown by others for LAMP1[64,65]. However, Dynasore inhibited endocytosis of HIV by T cells suggesting that clathrin-dynamin endocytosis is driving HIV uptake in the endosome as shown in other studies[11,65–67]. HIV could be transiently passing through EEA1+ endosomes and rapidly degraded in LAMP1+ endolysosomes or possibly trafficked through other, less well-characterized

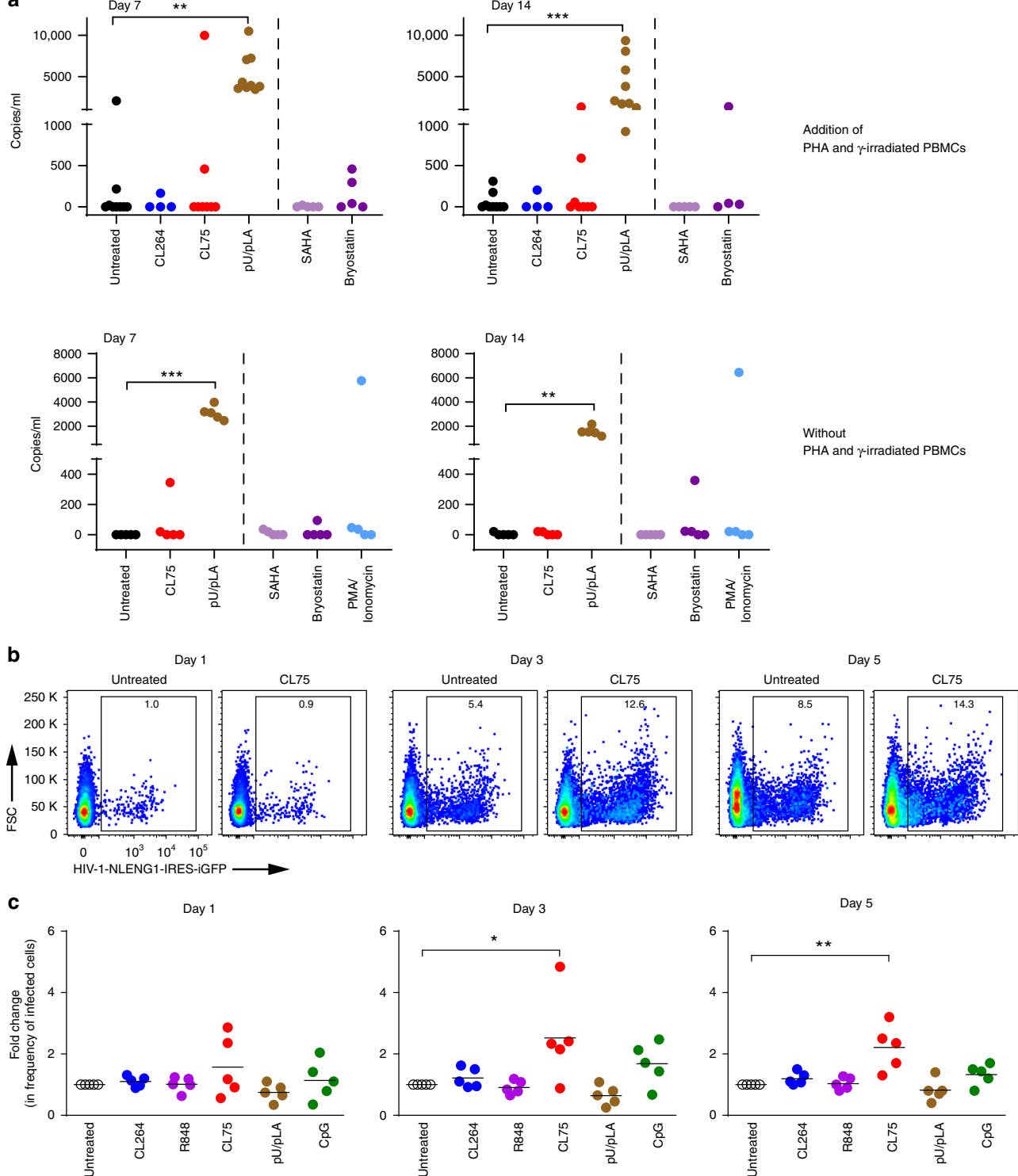

**Fig. 7 TLR8 increases HIV replication and reverses latency in CD4+ T cells. a** Viral outgrowth assay: CD4+ T cells isolated from aviremic patients were activated with γ-irradiated PBMCs and PHA (nine donors) or left untreated (five donors) and stimulated with 5 μg/ml of CL264 (TLR7), CL75, or pU/pLA (TLR8) for the time points indicated. Cells from four treated and all five untreated donors were stimulated with SAHA (335 nM), Bryostatin (10 nM), or PMA/ionomycin (500 nM/1.3 μM) for 16 h. HIV RNA was quantified in the supernatant by qPCR. **b**, **c** CD4+ T cells isolated from healthy blood donors were TCR-activated in the presence or absence of 5 μg/ml CL264, R848 (TLR7/8), CL75, pU/pLA, or CpG (TLR9, 5 μM) for 48 h prior to infection with cell-free NLENG1-IRES (X4 HIV-1). The frequency of eGFP+ T cells over time was quantified using flow cytometry; representative dot plot examples of untreated and CL75-treated CD4+ T cells 1–5 days post HIV infection are shown in **b**. Quantification in **c** is presented as fold change relative to HIV-1-infected cells in the absence of TLR ligands. The mean value of five independent experiment is indicated. Statistical significance was determined using repeated measures mixed effects model with Dunnett's post-test on log-transformed data; significance levels: *p < 0.05; **p < 0.01; ***p < 0.001. Source data are provided as a Source Data File.

endosomal compartments in the T cell. Whether TLR8 signaling originates from this compartment or if the viral RNA is trafficked further is currently unclear. Acidic environment is not a requirement for TLR8 activation. Endosomal TLR8 dimers are cleaved by neutral proteases, and studies in myeloid DCs have shown that retaining HIV or TLR8 ligands in early endosomes by preventing maturation, increases activation[34]. HIV endocytosis was not completely inhibited in our experiments. Dynasore has been shown to interfere with HIV replication, therefore target CD4+ T cells were only pre-treated with Dynasore and washed prior to infection (co-culture with HIV donor cells)[68]. This partial effect can be due to reversible inhibition of dynamin by Dynasore and its loss of potency due to the presence of serum in the culture media[69].

The adjuvant contribution of TLR8 engagement to cytokine production ranged from nothing (IL-2, IL-4, TNFα) to 2-50 fold, with IL-6 and IL-1β being the highest. TCR activation by itself efficiently induces IFN-γ, resulting in a modest adjuvant effect from TLR8 stimulation. This was also evident in the HIV-mediated response. However, both X4 and R5 HIV significantly increased IL-6 and IL-17 production in a TLR8-dependent manner, suggesting TLR8 signaling pathways more strongly induce these cytokines. The TLR7 ligand CL264 induced lower levels of IL-6 than TLR8 ligands while the TLR9 ligand CpG showed no effect or even decreased cytokine secretion induced by TCR activation of the T cells. CL264 is a synthetic TLR7 ligand (adenine analog) and we cannot guarantee that it does not show some cross reactivity to TLR8[70]. The TLR8 inhibitors CU-CPT9a and CU-CPT9b successfully inhibited the cytokine responses to TLR8 ligands and to endosomal HIV, suggesting HIV ssRNA is engaging TLR8.

In a recent study by Dominguez-Villar et al.[42], HIV-1 was reported to induce TLR7-mediated anergy in CD4+ T cells, promoting viral replication[42]. In this study, the TLR7 ligands Imiquimod (R837) and CL264 inhibited secretion of IL-2, IL-4, IFN-γ, and IL-17 in CD4+ T cells. This is in contrast to our findings that R837 and CL264 rather induced IL-6 secretion—which was not measured in the study by Dominguez-Villar. However, we did see inhibition of T cell cytokine production and upregulation of activation markers with the TLR9 ligand, CpG, although CpG was not investigated in the study by Dominguez-Villar et al.[42]. In addition, Dominguez-Villar et al. show increased viral replication in response to R837, which is different from our findings using CL264 that did not affect viral replication. We cannot explain these discrepancies, but the ligand dose, timing, and experimental assays could play a role. Moreover, Dominguez-Villar et al. used a model of productive HIV-1 infection with cell-free virus, and it is unclear how cytoplasmic HIV-1 RNA would enter the endosome and efficiently engage TLR7 in this infection model since the virus would fuse with host cell membranes and enter the cytosol[42]. Therefore, more research is needed to fully understand the mechanism by which TLR7 is activated by cytoplasmic HIV RNA. TLR2 and TLR5 have also been implicated in modulating TCR responses in CD4+ T cells[36,71] and we observed cytokine secretion induced by the TLR2 ligand FSL-1 in our study. HIV-1 may express TLR2 ligands; however, since TLR8-specific inhibitors dose-dependently inhibited HIV-1-induced cytokine responses and did not affect FSL-1-induced cytokines, we feel confident that TLR8 is the major receptor engaged by HIV-1 in our study. Other immune cells expressing TLR8 will respond to endosomal HIV ssRNA with cytokine secretion, and in particular myeloid cells such as macrophages and DCs. Whereas plasmacytoid DCs sense endosomal HIV-1 ssRNA by TLR7 and induce protective type I interferon responses[31,72], myeloid DCs, and macrophages sense endosomal HIV-1 ssRNA by TLR8 but fail to induce a protective immune response (e.g.

type I IFNs)[33–35]. In the T cell zone of the lymph node, the contribution of TLR8-mediated cytokine from a large number of T cells could play a role in HIV pathophysiology. IL-6 plays a central role in HIV infection. It promotes Th17 and inhibits Treg differentiation, and acts as a co-stimulatory factor enhancing proliferation and cell survival by counteracting activation-induced cell death[73]. A recent study shows that IL-6 is a stronger predictor of fatal events than C-reactive protein in ART-treated patients[74].

Synthetic TLR ligands are efficient activators of TLRs. However, differences in their structure can affect the level of activation. The natural ligands for TLR7 and 8 are ssRNAs in the forms of degradation products, nucleosides (guanosine and uridine, respectively), and oligoribonucleotides engaging different binding pockets. CL75, CL264, and R837 are synthetic imidazoquinolinone derivatives (IQDs) that are able to strongly interact with TLR7 and 8 homodimers and activate signaling in the absence of oligonucleotides, which are believed to function mainly to enhance the binding affinities of the nucleosides/IQDs[70,75–77]. PolyU is solely capable of activating TLR8, but interactions are stronger if combined with degradation products (uridine, endogenous nucleoside derivatives or IQDs). In addition, the structure of ligands affects their solubility; polyU is membrane impermeable and requires endocytosis whereas IQDs are lipid soluble and could engage TLR8 in compartments different from polyU. Both polyU and CL75 are considered more specific activators of TLR8 than TLR7. However, structural binding studies show that some cross-specificity is possible[70]. Although differences in potency was expected, we were surprised by the consistent differences observed between CL75 and polyU: polyU was most efficient in HIV reactivation and in upregulation of T cell activation markers whereas CL75 was most potent in enhancing HIV replication. Other studies have also pointed out differences in signaling between different TLR7/8 ligands[78]. Human PBMCs respond to R848 and ssRNA with pro-inflammatory cytokines but only ssRNA was able to induce type I IFNs[79] and recent studies suggest ligand-induced trafficking of TLR containing endosomes dictates different responses[22,80,81].

Despite the success of ART in reducing viremia to undetectable levels, virus eradication has remained elusive. Since ART is only effective against actively replicating HIV-1, long-lived latently infected cells are not eliminated and, when activated, are capable of reseeding the infection in the absence of ART. Our data show that TLR8 stimulation boosts the activation of latent HIV-1. PolyU's latency reversal activity surpassed that of other better studied latency reversing agents (LRAs) such as SAHA and Bryostatin. The observed lack of activity of SAHA is in agreement with a recent study demonstrating that LRAs that do not activate T cells do not induce substantial increase in intracellular HIV-1 mRNA in patient cells[82]. The latency reversal activity of PolyU is most likely mediated by NF-κB as it is a well-known transcriptional regulator of the HIV promoter[83]. In addition, TLR8 activation induces IL-6 secretion that can promote the expression of NFATc2 which in turn enhances HIV transcription by binding to the HIV LTR[84,85]. *Trans*-infection of latently infected CD4+ T cells could thus result in reactivation of HIV-1 by activation of the TCR with concomitant TLR8 signaling from endosomal degradation of virus particles. Even well-controlled HIV patients on ART experience blips of viral replication, most likely from activation of reservoirs[86,87]. This is one among many suggested explanations for why long-term HIV patients suffer from low-grade chronic inflammation and premature aging[2,3]. TLR8 boosting of TCR cytokine responses and viral replication could thus contribute to sustained inflammation in HIV, although this would have to be tested in in vivo models of HIV infection.

It has been suggested that reactivation of latent virus followed by an intensive course of ART can represent a therapeutic strategy to eradicate the virus. Several compounds including TLR agonists have been proposed as candidates with conflicting results[88]. In recent studies, TLR7 agonist GS-9620 induced pDC-mediated reactivation of latent HIV in PBMCs isolated from HIV patients on ART and delayed viral rebound in simian-human immunodeficiency virus (SHIV)-infected monkeys when administered with broadly neutralizing antibodies[89,90]. In another study, ligands for all human TLRs were tested for their efficacy to reactivate latent virus[91]. Only high doses of the TLR2/1 ligand Pam$_3$CSK$_4$ induced HIV-1 replication in latently infected central memory T cells. In an earlier study, the TLR5 ligand flagellin, but not Pam$_3$CSK$_4$, was able to reverse latency[91,92]. Synthetic TLR8 ligands have been reported to reverse latency in PBMCs, monocytic cell lines, and T cell lines[93–95]. Here we demonstrate the direct effect of TLR8 stimulation in highly pure primary human CD4+ T cells on activation and production on inflammatory cytokines. We also show that TLR8 stimulation contributes to reversion of latency in patient-derived latently infected CD4+ T cells, thus adding knowledge to a growing body of evidence that synthetic TLR ligands can play an important role in HIV cure research. These results reinforce the clinical relevance of our study and suggest TLR8 agonists should be considered for "shock-and-kill" strategies alongside LRAs already in clinical trials.

One major obstacle for studying TLRs in isolated primary human CD4+ T cells is the possible presence of contaminant cells such as monocytes and DCs that might also respond to TLR8 stimulation. In comparison to negative isolation, positive isolation by magnetic beads or FACS sorting yield higher purity (>99%) but can alter CD4+ T cell functional responses due to antibody labeling, and the physical stress applied to the cells during FACS sorting. We therefore used a negative isolation protocol in our study with which we consistently achieved CD3+CD4+ T cell purities of >95% with only minimal contamination of CD14+ (0.18%) cells with a monocyte phenotype or CD303+ (0.14%) or CD123+ (0.24%) cells with a DC phenotype. In addition, we confirmed IL-6 induction by TLR8 ligands in positively isolated CD4+ T cells, where we achieved a purity of >99.5% CD3+CD4+ T cells. Finally, TLR8 ligand induced IL-17 and IFN-γ was assessed by intracellular staining and flow cytometry and cytokine responses were inhibited by TLR8 inhibitors, further confirming that CD4+ T cells express functional TLR8.

Only a few other studies have investigated the direct effect of TLR8 stimulation on CD4+ T cell functions. TLR8 stimulation with poly-G reversed the ability of regulatory T cells to suppress T cell proliferation[40], whereas stimulation with the TLR7/8 ligand R848 in conjunction with TCR activation enhanced proliferation, IFN-γ, IL-8, and IL-10 secretion in memory CD4+ T cells[39]. A recent study reported TLR8-mediated reversal of Treg suppressive functions through inhibition of glucose uptake and glycolysis[96]. Our data show that TLR8 and TLR7 stimulation enhanced TCR activation by increasing the level of phosphorylated proteins involved in TCR signaling such as ribosomal protein S6 and, MAPK signaling (p38, ERK1/2) and NFκB p65. Unfortunately, due to the lack of antibodies that work satisfactory in phospho-flow, we could not assess TLR-specific signaling (IRAK4, TRAF6, IRFs). Despite comparable activation of TCR downstream signaling pathways, TLR8 stimulation was sufficient to induce IL-6 production from CD4+ T cells by itself and potently increased TCR activation, whereas the TLR7 ligand CL264 was inefficient and required additional TCR stimulation to increase low-level IL-6 secretion. Moreover, TLR7 stimulation showed no effect on IFN-γ and IL-17. Differences in potency of the ligands or in signaling pathways downstream of TLR7 and TLR8 seem plausible but needs to be proved.

Our data further suggest that TLR8 signaling in CD4+ T cells favors differentiation towards the pro-inflammatory Th1/Th17

CD4+ effector T cell axis. Mechanistically this is most likely driven by TLR8-induced IFNγ and IL-6 in a paracrine/autocrine feedback loop: IFNγ is known to induce T-bet (STAT1 signaling) and IL-6 induces RORγT (STAT3 signaling). Interestingly, IL-6 in this scenario is often said to originate from innate cells whereas we show that T cell intrinsic TLR8 activation can induce this feedback loop. Th1 and Th17 effector cells are proposed to support HIV-1 long-term persistence in patients receiving ART[49], further supporting a pro-viral role of TLR8 stimulation in T cells. TLR ligands have been investigated for the ability to act as adjuvants to modulate the vaccine immune response, primarily due to their action on innate immune cells[97]. Our results show that stimulation of CD4+T cells with TLR8 ligands even in the absence of innate immune cells increases the frequencies of IFN-γ- and IL-17-producing effector cells. The observation that TLR8 ligands show a direct effect on CD4+ T cells may thus have implications for vaccine research.

Based on previous studies and our data, we propose a model where a "dead-end" pathway for HIV-1 infection can contribute to viral dissemination: HIV-1-infected APCs or T cells migrate to the lymph node and transmit the virus to uninfected CD4+ T cells during cell-to-cell contact. If the appropriate co-receptor is present, the virus can fuse from the endosome and initiate productive infection by replicating in the target cell. Conversely, endosomal HIV-1 that is not capable of undergoing fusion due to the absence of the appropriate co-receptor can be degraded and engage TLR8. In our in vitro model CXCR4/CCR5 inhibitors were used to trap HIV-1 in endosomal compartments, thus optimizing the conditions to fully induce TLR8 activation. The extent of co-receptor mismatch will vary in HIV-infected individuals and over time within the same patient, making it difficult to predict to what extent T cell intrinsic TLR8 engagement will influence HIV pathology in patients. The relevance should be most apparent in secondary lymphoid tissues where the majority of HIV replication occurs due to the presence of high number of T cells. Secretion of inflammatory cytokines in the extracellular milieu may in turn act on productively infected cells, enhancing viral replication and commit the T cells to differentiation towards the pro-inflammatory Th1/Th17 axis and thus contribute to the pathology of HIV infection such as viral dissemination and the establishment of long-lived HIV reservoirs. Moreover, TLR8 activation by endosomal HIV may support the low-level viral replication and "blips" occurring in patients on ART, and thus contribute to low-grade chronic inflammation in these patients. More studies are needed to assess the contribution of cell-to-cell transmission and TLR8 activation of CD4+ T cells to HIV disease in humans.

## Methods

**General**. Supplementary Table 2 lists all antibodies with dilutions at which they were used for confocal microscopy, western blot, flow cytometry, cell stimulation, and cell depletion in the study. Gating strategies for flow cytometric analysis of cytokine production, protein phosphorylation, surface marker, and transcription factor expression as well as HIV-1 infection of CD4+ T cells are exemplified in Supplementary Fig. 7.

**Cells**. Buffy coats from healthy blood donors were provided by the Blood Bank (St Olav's Hospital, Trondheim) with approval by the Regional Committee for Medical and Health Research Ethics (REC Central, Norway, No. 2009/2245). Informed consent is routinely asked for by the Blood Bank and buffy coats are made available for research when consent is obtained. PBMCs were isolated using density gradient centrifugation (Lymphoprep, Axis-shield PoC). Highly pure CD4+ T cells were isolated from PBMCs by two different methods using magnetic beads. Negative isolation using the CD4+ T Cell Isolation Kit (Miltenyi Biotec) is based on depletion of non-CD4+ cells and leaves CD4+ T cells "untouched". Unless stated otherwise, this was the method used throughout the study. Positive isolation was carried out for select experiments using Dynabeads CD4 Positive Isolation Kit (Thermo Fisher Scientific). CD14+ monocytes isolated with CD14 Microbeads (Miltenyi Biotec) from the CD4-negative fraction were used in indicated

experiments to study the effect of contaminating cells. CD4+ T cell purity for all isolations was assessed by flow cytometry using anti-CD4 Alexa 700 (eBioscience) and anti-CD3 Brilliant Violet (BV) 785 (BioLegend) antibody staining. Frequencies of contaminating cell populations were identified by antibody staining for CD8 (BV605, BioLegend), CD14 (PE/Cy7, eBioscience), CD19 (PE, eBioscience), CD11c (PE, eBioscience), CD16 (FITC, BD Biosciences), CD303 (APC, Miltenyi Biotec), CD123 (PE, Miltenyi Biotec), HLA-DR (BV785, BioLegend), and CD80 (FITC, BD Biosciences). Data were acquired on a BD LSRII flow cytometer and analyzed using FlowJo software (FlowJo, LLC). CD4+ T cell purity was on average >95% for negative isolation and >99% for positive isolation (Fig. S2, gating strategies in Supplementary Fig. 7). Only isolations with at least 94% CD4+ T cell purity were used for experiments. CD4+ T cells were cultured in RPMI 1640 (Sigma-Aldrich) supplemented with 10% pooled human serum (The Blood Bank, St Olav's Hospital, Trondheim, Norway) or fetal bovine serum (FBS) (Thermo Fisher Scientific) for HIV infection experiments. MDMs were generated by adherence on FBS pre-treated glass slides placed in 24-well plates. The cells were left to adhere for 1 h in RPMI 1640 supplemented with 10% pooled human serum at 37 °C and 5% $CO_2$. The attached monocytes were then washed three times with Hanks's Balanced Salt Solution (GIBCO) and cultured for 6 days in RPMI 1640 supplemented with 10% pooled human serum and 10 ng/ml recombinant M-CSF (R&D Systems). At day 6 the media was replaced with M-CSF-free media. HeLa cells (ATCC) and HEK293T (ATCC) cells were cultured in DMEM (Sigma-Aldrich), supplemented with 10% FBS. SupT1 human T lymphoblast cells (ATCC) were cultured in RPMI 1640 (ATCC modification) (Gibco) supplemented with 10% FBS and penicillin/strep-tomycin (Thermo Fisher Scientific).

**CD4+ T cell activation and synthetic TLR ligand stimulation**. Different methods were used for TCR activation of T cells. For HIV-1 cell-to-cell transmission using HeLa cells (imaging), HIV-1 in vitro infection, or stimulation with synthetic TLR ligands, CD4+ T cells were TCR activated in anti-CD3-coated plates (clone OKT3, eBioscience, 5 µg/ml, 1 h) in the presence of 1 µg/ml anti-CD28 (clone CD28.2, eBioscience). In indicated synthetic ligand stimulation experiments and HIV-1 cell-to-cell transmission using HEK293T cells (cytokine analysis), CD4+ T cells were activated with activation beads (Dynabeads Human T-Activator CD3/CD28, Thermo Fisher Scientific) according to the manufacturers' protocol for the time indicated. In phospho-flow signaling experiments, CD4+ T cells were incubated with biotinylated anti-CD3 (1 µg/ml) and anti-CD28 (0.5 µg/ml, both eBioscience) for 2 min, before cross-linking with avidin (50 µg/ml, Zymed, $t = 0$) to achieve simultaneous activation of TCR signaling. For CXCR4 and CCR5 expression analysis, CD4+ T cells were activated with 0.75 µg/ml PHA (Sigma-Aldrich) and stained at indicated timepoints for CXCR4 (PE/Cy5, BioLegend) and CCR5 (PE/cy7, BioLegend) expression. For viral outgrowth assay, healthy donors (lymphoblasts) and HIV patient CD4+ T cells were activated with 0.75 µg/ml PHA.

For TLR stimulation, the following synthetic ligands were used: 5 µg/ml of FSL-1 (TLR2), CL264 (TLR7), R837 (TLR7), R848 (TLR7/8), CL75 (TLR8), polyU (TLR8) complexed with poly-L-Arginine (Sigma-Aldrich, 5 µg/ml, pU/pLA) to facilitate uptake, TL8-506 (TLR8), or 5 µM of CpG-ODN 2006 (TLR9), and GpC-ODN 2006 (TLR9 control). All TLR ligands were purchased from Invivogen. Dose response analysis was carried out using increasing concentrations 0.2–5 µg/ml of TLR2/7/8 ligands or 0.2–5 µM for CpG.

**TLR8 and endocytosis inhibition**. CD4+ T cells ($0.5 \times 10^6$) were cultured in 96-well plates and pre-treated for 2 h with the TLR8-specific inhibitors CU-CPT9a, CU-CPT9b, the negative control compound CU-CPT6 or DMSO at the indicated concentration in 100 µl media before TCR activation and stimulation with synthetic TLR ligands. Supernatants from the cells were collected for cytokine analysis or cells were analyzed by intracellular flow cytometry for cytokine production. For HIV-1 cell-to-cell transmission experiments, CD4+ T cells were pre-treated with the CXCR4 or the CCR5 receptor antagonist for 24 h as described below. T cells were then treated with 5 or 25 µM TLR8-specific inhibitors for 2 h before co-culture with HIV-1 producing HEK293T cells and TCR activation. Supernatants from the cells were collected for cytokine analysis after 24 h. For inhibition of endocytosis, CD4+ T cells were pre-treated with Dynasore (80 µM, Sigma) for 1 h, washed with PBS, and co-cultured with HEK293T cells for 24 h. Cells were then trypsinized, washed, and fixed with 4% PFA before analysis by microscopy-assisted flow cytometry (FlowSight, Amnis). The data were analyzed using IDEAS Software (Amnis).

**HIV-1**. HIV-1-Gag-iGFP and HIV-1-Gag-iGFP_JRFL: full-length molecular clone of HIV derived from pNL4-3 with green fluorescent protein (GFP) inserted into the Gag protein between the MA and CA domains of Gag, with HIV protease cleavage sites created to flank the GFP insertion. In HIV-1-Gag-iGFP_JRFL a fragment of the Env gene from JRFL is cloned into the place of the NL4-3 Env (NdeI to BamHI) making the virus R5-tropic. The plasmids were obtained through the NIH AIDS Reagent Program, Division of AIDS, NIAID, NIH: HIV-1-Gag-iGFP and HIV-Gag-iGFP_JRFL from Dr. Benjamin Chen[6,98]. NLENG1-IRES: full-length molecular clone of HIV derived from pNL4-3 carries enhanced green fluorescent protein (eGFP) inserted in front of an IRES sequence and Nef[12,99]. The

plasmid was a kind gift from Dr. David N. Levy (NYU College of Dentistry, USA). For preparation of virus stocks, HEK293T cells were cultured in T75 flasks in the absence of antibiotics for 24 h. Cells were transfected with 10 µg of pNLENG1-IRES, HIV-1-Gag-iGFP, or HIV-1-Gag-iGFP_JRFL using Gene Juice (EMD Millipore) according to the manufacturer's protocol. The media was changed the following day. The virus-containing supernatant was collected 48 and 72 h post transfection and cleared from virus debris by centrifugation. To concentrate the virus stocks, Lenti-X concentrator (Clontech) was used according to the manufacturer's protocol. The virus pellet was resuspended in 1/100 of the original volume and stored in −80 °C. Virus titer was determined by HIV-1 p24 ELISA assay (Express Bio).

**HIV-1 infection**. For in vitro infection CD4+ T cells, the cells were TCR-activated in the presence or absence of TLR ligands. NLENG1-IRES or HIV-1-Gag-iGFP_JRFL (400 ng P24) was added to $10^6$ stimulated CD4+ T cells in 500 µl of RPMI 1640 supplemented with 10% FBS and 10 µg/ml polybrene (Santa Cruz Biotechnology). Spinofection was carried out by centrifugation (1200 g) for 2 h at 20–25 °C. The supernatant was aspirated, and CD4+ T cells were cultured in RPMI 1640 supplemented with 10% FBS and 100 U/ml IL-2 (R&D systems) in the presence or absence of 5 µg/ml of the CXCR4 antagonist AMD3100 (Sigma-Aldrich) and TLR ligands. Productive infection was assessed at various time points by flow cytometry after fixation with 4% PFA for 10 min. Data were acquired on a BD LSRII flow cytometer and analyzed using FlowJo software (FlowJo, LLC.).

For HIV-1 HeLa cell-to-T cell and HEK293T cell-to-T cell transmission, HeLa cells were cultured for 24 h in 12-well plates for flow microscopy or glass bottom 96-well plates for confocal imaging, or HEK293T cells were cultured in 96-well polystyrene plates for cytokine analyses. The HeLa or HEK293T cells were then transfected for 48 h with 1 µg (12-well plate) or 0.12 µg (96-well plate) of pNL4-3 Gag-iGFP using Gene Juice (EMD Millipore) according to the manufacturer's protocol. For T cell-to-cell transmission, donor CD4+ T cells were TCR-activated and spinofected with HIV-1-Gag-iGFP (400 ng P24) as described above. For MDMs-T cell transmission, $2 \times 10^6$ MDMs were infected HIV-1-Gag-iGFP (400 ng P24) in 120 µl of RPMI 1640 supplemented with 10% FBS and 10 µg/ml polybrene. Spinofection was carried out by centrifugation (1200 g) for 2 h at 20–25 °C. The media was replaced, and the cells were cultured for 6 days. Productive infection was assessed by epifluorescence microscopy prior to co-culture. For imaging experiments, acceptor CD4+ T cells were TCR activated for 48 h as described above and stained with Cell Tracker Deep Red Dye (Thermo Fisher Scientific) before combining with HIV-1 producing HeLa cells, HEK293T cells, donor CD4+ T cells or MDMs. For SupT1 cell-to-T cell transmission, $10^6$ SupT1 cells were spinofected with NLENG1-IRES HIV-1 (600 ng P24). Following a 48-h incubation, productively infected SupT1 cells were co-cultured separately or together in a 24-well plate with 0.4 µm trans-well permeable supports (Costar) with TCR-activated CD4+ T cells stained with Cell Tracker Deep Red Dye (Thermo Fisher Scientific). Productive infection of CD4+ T cells were assessed by flow cytometry after fixation with 4% PFA for 10 min. Data were acquired on a BD LSRII flow cytometer and analyzed using FlowJo software (FlowJo, LLC.).

For cytokine analysis experiments, CD4+ T cells were TCR activated as described above at the time of co-culture with HIV-1 producing HEK293T cells. In some experiments, CD4+ T cells were pre-treated with the CXCR4 receptor antagonist AMD3100 (5 µg/ml, Sigma-Aldrich) or the CCR5 receptor antagonist Maraviroc (5 µg/ml, Sigma-Aldrich) for 24 h prior to co-culture. Treatment with the receptor antagonists was maintained throughout the experiments. Supernatants from the cells were collected for cytokine analysis at indicated timepoints.

**Confocal imaging and microscopy-assisted flow cytometry**. For confocal imaging, CD4+ T cells were collected from the co-culture, trypsinized for 5 min at 37 °C with 0.25% Trypsin/EDTA (Sigma-Aldrich) to remove surface-bound virus and fixed with 4% PFA. Immunostaining was performed with anti-LAMP1 (Santa Cruz H4A3 or Abcam 24170), anti-EEA1 (Santa Cruz H-300) and anti-p24 (Abcam 9071) primary antibodies and Alexa555 or 405-conjugated secondary antibodies (Thermo Fisher). Briefly, cells were blocked and permeabilized in PBS + 1% BSA + 0.05% saponin for 1 h before primary antibodies were added (1:100) and incubated at 4 °C over night. The next day cells were washed three times PBS+BSA+saponin and stained with secondary antibody (2 µg/ml) for 1 h at room temperature. Finally, the cells were washed three times and resuspended in 100 µl PBS+1% BSA (protein was included in the washing buffer to avoid loss of cells). Ten microliters cell suspension were added to a glass bottom 96-well plate (Cellvis) and spun down at 600 g for 5 min, before 90 µl glycerol was added and cells spun down again at 600 g for 10 min. Imaging was performed on an inverted Leica SP8 or Zeiss LSM 880 confocal microscope, using ×40 or ×63 oil immersion objectives. For imaging of virological synapses, the remaining HeLa cells/MDMs and CD4+ T cells attached to the glass bottom plate or glass slide were fixed with 4% PFA before imaging of HIV-1-Gag-iGFP and Cell tracker deep red-stained acceptor CD4+ T cells. For flow microscopy analysis, CD4+ T cells were collected from co-culture with HeLa cells, washed with PBS (Sigma-Aldrich), and cultured for the indicated time points. Cells were then trypsinized, washed, and fixed with 4% PFA before analysis by microscopy-assisted flow cytometry (FlowSight, Amnis) and BD LSRII flow cytometry. The data were analyzed using IDEAS Software (Amnis) and FlowJo software (FlowJo, LLC). Quantification of the number of

trypsin-resistant HIV puncta per cell (Fig. 1b) was done by counting 14 random z-stacks containing 2–5 T cells per stack (a total of $n = 61$ cells), and the average number of puncta per cell in each stack is plotted. Counting was done manually on max intensity projections using the multi-point tool in Fiji.

**Immunoblotting.** Cells were lysed in RIPA buffer supplemented with 4 M urea and cOmplete Protease Inhibitor Cocktail (Merck). Protein lysates were quantified using Bradford Protein Assay (Bio-Rad) and run on 10% Nupage minigels (Invitrogen), then transferred to a nitrocellulose membrane using Invitrogen's iBlot system. After blocking for 1 h with BSA, the membrane was incubated with anti-TLR8 Rabbit mAb (D3Z6J, Cell Signaling Technology) at 4 °C for 24–48 h. After washing with TBS-T, the membrane was incubated with polyclonal goat anti-rabbit HRP (Dako) for 1 h. The blots were developed with the SuperSignal West Femto (Thermo Scientific) and visualized with Odyssey Fc imaging system (LI-COR). Image Studio analysis software was used for band intensity quantification. Uncropped and unprocessed scans of the blots are in the Source Data file.

**Phospho-flow cytometric signaling analysis.** CD4+ T cells were seeded at a density of $5 \times 10^7$ cells/ml and rested for 15 min at 37 °C. TCR, TLR7 (CL264, 5 µg/ml), and TLR8 (CL75, 5 µg/ml) signaling was simultaneously induced for 0–30 min as described above. Cells were fixed with BD Phosphoflow Fix Buffer I (BD Biosciences) and barcoded in two dimensions with five dilutions of Pacific Blue and Alexa Fluor 488 (both Molecular Probes, Invitrogen, Carlsbad, CA, USA). After permeabilization with BD Phosphoflow Perm Buffer III (BD Biosciences), cells were stored at −80 °C overnight. Cells were stained with anti-CD4 (Alexa Fluor 700, eBioscience) and phospho-epitope-specific antibodies. Alexa Fluor 647-labeled antibodies to CD3ζ (pY142), ZAP70 (pY319), SLP76 (pY128), NF-κB p65 (pS529), and STAT3 (pY705) were purchased from BD Biosciences; Alexa Fluor 647-conjugated antibodies to Akt (pS473), S6 Ribosomal Protein (pS235/pS236), NF-κB p65 (pS536), p44/42 MAPK (Erk1/2IκBT202/pY204), and p38 MAPK (pT180/pY182) were purchased from Cell Signaling Technology; eFluor 660-labeled antibody to IκB alpha (pS32/pS36) was purchased from eBioscience. Phospho-flow cytometric data were acquired on a BD LSR Fortessa flow cytometer and analyzed using Cytobank software (Cytobank, Inc.) as exemplified in Supplementary Fig. 7. The level of phosphorylation was analyzed as arcsinh ratio of median fluorescence intensity for the various signaling proteins, calculated by normalizing against unstimulated CD4+ T cells as a control.

**Flow cytometry.** CD4+ T cells ($2.5 \times 10^6$/ml) were activated in the presence or absence of synthetic TLR ligands for the indicated periods as described above. For analysis of activation marker expression, cells were stained with anti-CD25 (Brilliant Violet (BV) 510), anti-CD40L (FITC), anti-CD69 (PE), anti-CD80 (PE/Cy7), anti-HLA-DR (BV785), and anti-PD-1 (BV605, all BioLegend). Production of intracellular effector cytokines was analyzed at different time points after activation and on day 8 upon short-term re-stimulation (Cell Stimulation Cocktail, eBioscience, 6 h). Protein transport inhibitor cocktail (eBioscience) was added for the last 4 h before harvest of the cells. Staining with Fixable Viability Dye eFluor 780 (eBioscience) and fluorescent antibodies to CD3 (FITC or BV785, BioLegend) and CD4 (Alexa Fluor 700, eBioscience) was performed before cells were fixed and permeabilized (FOXP3 buffer set, BD Biosciences). In indicated experiments cells were additionally stained for CD45RO (PerCP/Cy5.5, BioLegend) and CD45RA (BV785, BioLegend) before fixation. Intracellular cytokine and transcription factor staining was performed with fluorescent antibodies to IFN-γ (FITC, Miltenyi Biotec or PE, eBioscience), IL-4 (PE-Vio615, Miltenyi Biotec), IL-17 (BV510, BioLegend), TNF-α (BV421, BioLegend), IL-2 (PE/Cy7, eBioscience), T-bet (PE, eBioscience), and RORγt (APC, eBioscience). Data were acquired on a BD LSRII flow cytometer and analyzed with FlowJo software (FlowJo, LLC) as exemplified in Supplementary Fig. 7.

**Analysis of secreted cytokines.** In stimulation experiment with synthetic TLR ligands, CD4+ T cells were TCR activated in the presence or absence of TLR ligands as described above. In HIV cell–cell transmission infection experiments, CD4+ T cells were co-cultured with HIV-1-transfected HEK293T as described above. Supernatants collected at indicated time points were analyzed by ELISA using Human IL-6 DuoSet ELISA (R&D Systems) or by multiplex analysis using the ProcartaPlex Human Cytokine and Chemokine panel (Affimetrix, eBioscience) according to the manufacturer's protocol.

**Patient cohort.** Nine HIV-1-infected patients were recruited from the St. Olav's infectious disease outpatient clinic. The participants were on ART and had suppression of plasma HIV-1 viremia below the limit of detection (<20 copies HIV-1 RNA/ml). Patient 7 had 119 copies/ml at the time of blood acquisition. The viral load test was repeated 2 months later, and the viral load was 33 copies/ml without change of regimen. None of the patients had other chronic inflammatory diseases or used medication affecting inflammation. All participants provided written informed consent to participate in the study. This study was approved by the Regional Committee for Medical and Health Research Ethics (REK 2014/1507). Patients characteristics are summarized in Supplementary Table 1.

**Viral outgrowth assay.** PBMCs from HIV-1-infected patients were isolated using density gradient centrifugation (Lymphoprep, Axis-shield PoC). CD4+ T cells were enriched by negative isolation (CD4+ T Cell Isolation kit, Miltenyi Biotec). Activated CD4+ T cells were depleted by labeling with biotin-conjugated anti-CD69, anti-CD25, and anti-HLA-DR (all eBioscience) and magnetic separation using anti-Biotin Microbeads and MS columns (both Miltenyi Biotec). Purified resting CD4+ T cells were stained with anti-CD4 (Alexa Fluor 700, eBioscience), anti-CD69 (PE, Biolegend), anti-CD25 (BV510, Biolegend), and anti-HLA-DR (BV785, Biolegend) and purity was assessed by flow cytometry using BD LSRII flow cytometer (BD Biosciences) and FlowJo software (FlowJo. LLC). Resting CD4+ T cells from patients ($0.2 \times 10^6$) were co-cultured with allogeneic γ-irradiated (30 Gray) PBMCs ($1.25 \times 10^6$) from healthy donors in RPMI containing 10% FBS, PHA (Sigma-Aldrich), and 100 U/ml IL-2 (R&D systems) in the presence or absence of TLR ligands, SAHA (335 nM, Sigma-Aldrich), Bryostatin-1 (10 nM, Sigma-Aldrich), or cell stimulation cocktail (PMA 500 nM + ionomycin 1.3 µM, eBioscience). At day 2 and day 9, media was removed and purified CD4+ lymphoblasts from healthy donors were added. The concentrations of TLR ligands were maintained throughout the duration of the assay whereas SAHA, Bryostatin-1 and cell stimulation cocktail were removed the next day. Some patient resting CD4+ T cells were stimulated with TLR ligands in the absence of allogeneic γ-irradiated PBMCs and PHA. Supernatants were collected at day 7 and day 14 and HIV RNA was quantified using Cobas HIV-1 quantitative nucleic acid test for Cobas 6800 system (Roche).

**Data and statistical analysis.** Flow cytometry data were analyzed using FlowJo (FlowJo, LLC), Cytobank (Cytobank, Inc.), and IDEAS Software (Amnis). Other data and statistical analyses were performed using GraphPad Prism (GraphPad Software, Inc.). Normal distribution of data after log-transformation was assumed. Statistical significance was determined from log-transformed datasets or AUC calculations by repeated measures one-way, two-way ANOVA followed by Dunnett's post-test or paired $t$-test as stated in the figure legends; significance levels: *$p < 0.05$; **$p < 0.01$; ***$p < 0.001$.

**Reporting summary.** Further information on research design is available in the Nature Research Reporting Summary linked to this article.

## Data availability
All relevant data are available from the authors. The source data underlying Fig. 1c+d, 2a–c, 3a+b, 4b, 5c+d, 6, 7a+c and Supplementary Figs 1a+d, 2b–e, 3, 4, 5a–c, 6 are provided as a Source Data File. The proteomics data set used to plot Supplementary Fig. 3 was retrieved from Rieckmann et al., https://doi.org/10.1038/ni.3693 with all datasets deposited online to ProteomeXchange Consortium PXD004352. The small-molecule TLR8 inhibitors, CU-CPT9a and CU-CPT9b, must be obtained through an MTA.

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

## Acknowledgements

The authors would like to thank Birgitta Åsjø for help in establishing the HIV cultivation and infection assays, Nadra Nilsen and Jane Awuh for helpful discussions, Anne Marstad for experimental assistance, Kirsti Løseth for assistance in the BSL-3 facilities, and Silja Egilsdottir for virus RNA quantification. All imaging was performed at the Cellular & Molecular Imaging Core Facility at NTNU. The research was supported by grants from the Research Council of Norway (245663 and 223255 [to T.H.F.], 187615 [to K.T.], the Liaison Committee between NTNU and the Central Norway Regional Health Authority (90176000 [to T.H.F.], 46056834 [to M.H.]), the Regional Health Authority for South-Eastern Norway (2013074, 2017119 [to K.T.]), Stiftelsen Kristian Gerhard Jebsen [to K.T.], The National Natural Science Foundation of China (20151300651 and 20181371453 [to H.Y.]), and Beijing Outstanding Young Scientist program (BJJWZYJH01201910003013 [to H.Y.]).

## Author contributions

H.Z.M., M.H., and T.H.F. conceived and designed the study; H.Z.M., M.H., M.S.B., C.L., and L.R. performed the experiments; H.H.Y. and Z.H. are responsible for the development and preparation of the specific small-molecule TLR8 inhibitors, CU-CPT9a and CU-CPT9b; S.A.N. provided viral load analysis; J.L. and K.T. contributed to phospho-flow cytometry experiments; J.K.D. carried out patient enrollment and facilitated blood sample acquisition; H.Z.M., M.H., and T.H.F. wrote the paper; all co-authors contributed to scientific discussions of results and in reviewing/commenting on the paper.

## Competing interests

The authors declare no competing interests.
