## [Peer Review File · Nature Communications]

Reviewers' comments:

Reviewer #1 (Remarks to the Author):

very nice paper and interesting molecular aspects. I am convinced that the data in this paper add substantially to mechanistic insights in TLR8 signalling in CD4+T-cells and potential effects on HIV replication.

However, the data should be positioned in a bigger perspective. The lymphoid system is more than CD4+ T-cells alone, and the effects of other innate immune cells which are triggered by TLRs predominate or even overshadow completely the effects shown here.

Minor comments:

1)an additional control for the endosomal uptake of HIV would be the blocking of the endosomal uptake machinery and thereupon the lack of HIV in the endosomes.

2)please, elaborate on the differences of the TLR8 agonists used as they show substantial differences in their effects

3)please, provide statistics to fig 3b

Roberto F. Speck

Reviewer #2 (Remarks to the Author):

Stimulation of TLR8 in antigen presenting cells enhances Th1 and Tfh cell responses, but several studies have also described T cell-intrinsic functions of TLR8.

In this study, Meås et al. report that TLR8 activation in purified human CD4 T cells leads to enhanced T cell activation and pro-inflammatory cytokine production. They propose that endosomal uptake of HIV-1 by cell-to-cell transmission activates TLR8 in CD4 T cells, and they show that stimulation of latently infected CD4 T cells with TLR8 agonists can reverse latency in vitro.

While the study addresses an interesting aspect of host-virus interaction, there are several major concerns.

The presented data do not sufficiently prove the claimed TLR8 stimulation by endocytosed HIV-1 in CD4 T cells. Along these lines, the statistical analysis of most of the data in the manuscript is insufficient, which severely compromises the conclusions.

Direct functions of TLR8 in naïve and memory CD4 T cells have been previously reported (Caron 2005, PMID:16034093), particularly in Tregs (Peng 2005, PMID:6123302, Li 2018, PMID:30344014). Here, the authors investigated the effects of TLR8 ligation in primary naïve CD4 T cells. Reversal of HIV-latency by TLR8 stimulation has also been previously demonstrated, albeit not by direct activation of TLR8 in T cells (Schlaepfer 2011, PMID:21357269).

Specific comments

1. The text claims that Fig. 1 and Fig S1 “confirm that HIV-1 is endocytosed (...) and routed to lysosomes for degradation”. However, Fig. 1B depicts a single LAMP1+/gag-GFP+ punctum. Adequate quantification of LAMP1+ HIV-1 containing endosomes is required to substantiate the claim.

2. Most figures lack sufficient statistical analysis or present data from single experiments.

a. Fig. 2A, Fig. 3B, Fig. 4, Fig. 5C lack statistical analysis.

b. Fig. 2A and Fig. S2C-E show individual values without SD or indication of replicates. Are these single replicates (n=1)?

c. Fig. S5A shows “one representative experiment repeated three times with similar results”, no statistical analysis.

d. Data in Fig. 2D, Fig 3A, Fig. 5D, Fig. 6 (Fig. 7?) were analyzed by student’s t-tests. However, t-tests are inappropriate when comparing more than two experimental groups.

e. Fig. S7 shows one experiment performed on pooled cells from four aviremic patients. It is unclear how many of the donors responded to the treatment.

3. TLR8-specific inhibitors were used to block HIV1-induced cytokine production (Fig. 6 /S6). While both inhibitors blocked CL075- and pU/pLA-induced cytokine production (Fig. 2D), only CU-CPT9b inhibited HIV-1-induced IL-6 production (and only when comparing log₂ fold change, not when comparing absolute values, Fig. S6). There seems to be little effect of TLR8 inhibition on HIV-induced IFN γ production, despite IFN γ being induced by TLR8 agonists and inhibited by TLR8 inhibitors (Fig. 3). IL-1 β production is potentially the most striking effect of TLR8 stimulation in CD4 T cells (Fig. 3A), yet, IL-1 β response to HIV-1 infection was not assessed. The proposed TLR8 stimulating effect of HIV-1 is not convincingly proven by the data in their current state. Genetic approaches may yield more definitive results, and several methods for gene editing of primary T cells have recently been published.

4. It is curious that pU/pLA reactivated viral replication in latently infected CD4 T cells from all donors, while CL075 did so only in one donor (Fig. 7A). In contrast, CL075 enhanced productive HIV infection, which pU/pLA did not (Fig. 7C). How do the authors explain these differences?

Minor comments

1. Residual content of 0.27% of CD14+ cells in the T cell preparations could potentially account for the observed IL-6 production in response to TLR ligands.

2. Fig. 2C shows the knock-down efficiency of siRNA transfected T cells. However, the only panel analyzing TLR8-silenced cells is in Suppl. Fig. 2D.

Reviewer #3 (Remarks to the Author):

Meas et al examines the expression and function of TLR8 in primary CD4 T cells. Furthermore, they explored the potential that HIV-1, through an endosomal route of entry can activate signaling through TLR8. Experiments include manipulating receptor function using a panel of TLR agonist and antagonist and assessing T cell function and HIV expression and replication. Major findings include showing that TLR8 is expressed in CD4+ T cells, that TLR ligands elicit T cell responses including the induction of cytokines and phenotypic markers that may reflect maturation of cells into TH1 and TH17 cells and that TLR8 can facilitate active HIV-1 replication and latency reversal. Strengths of the paper includes the exhaustive characterization of responses and the use of a library of TLR ligands and inhibitors to map out the role of TLR8. However, many of the TLR mediated responses are modest, the responses to ligands and inhibitors are highly variable, mechanisms of action are not explored and there are some concerns about the relevance of trans-infection models that are being used. Overall, the data are intriguing, they raise a number of relevant questions but the study is preliminary and requires controls and additional mechanistic studies. Some specific comments are below.

1. Fig 1 shows transfer of HIV to a target cell by an endosomal route. For this experiment HeLa cells are used to transfer HIV which raise some concerns about the relevance of the cell type. In addition, the results are not well described and appears that the process of entry is inefficient. For example, in fig 1B only one particle of four appears to be associated with endosome/lysosome. The number of particles associated with a synapse or a LAMP-1 compartment should be quantified since they are showing only selected images of a single or few cells. A minor point is that it seems these data should be presented with the other experiments that address HIV endosomal entry and TLR8 signaling (Fig 6 and 7).

2. Much of the paper focuses on TLR8 signaling and the potential that this influences CD4+ T cell function. Figure 2 addresses TLR8 expression with western blots and induction of IL-6. Actually, the IL-6 data should be presented in figures that includes induction of the other cytokines since that addresses function rather than expression. It would have been interesting to look at T cell subsets, such as memory cells, especially because of the importance of these cells in HIV latency. It was not clear as to the importance of the siRNA knock-downs since these cells and knockdowns were not used in the paper.

3. The cytokine expression data in Fig 2 and 3 should be combined. Other than IL-1b and IL-6 and maybe IFN γ , the induction of cytokines and markers are very modest, usually less than 2 fold. It would be helpful in appreciating the robustness of this response to include positive controls such as CD3+CD28, PHA or PMA.

4. Fig 4 the authors conclude that TLR8 acts in synergy with TCR. However, the cellular responses are modest and there are not statistics to support the conclusion of synergy. How is synergy being defined?

5. An interesting conclusions of the paper is that TLR8 signaling maybe biasing cell function towards TH1 cells and possibly TH17 cells, although the latter data are less robust. Expanding this observation and the mechanism driving this maturation would strengthen the paper.

6. The experiments examining whether endosomal entry of HIV trigger TLR8 signaling (Fig 6) are modest with the exception of IL-6. Furthermore, the variation in the activity of the different inhibitors was not discussed. Proviral levels should be measured to confirm that infection is not being established. Like HeLa cells, there are concerns about using HEK293 cells and it would be informative to confirm these results with macrophages or dendritic cells.

7. Experiments in Fig 7 suggest that TLR8 may influence HIV expression and latency reversal. A minor point is that technically they are measuring induction of RNA levels and not the outgrowth of virus, which is detected by expanding induced virus by infecting indicator CD4+ cells. It was somewhat surprising that PHA without TLR8 ligands did not induce HIV RNA from the ART samples. It would be informative to include other known latency reversing agents to get a sense of how efficient TLR8 ligands are at inducing HIV transcription and whether TLR8 agonist could synergize with these compounds. Proviral load should be measured since increase replication might reflect greater HIV infection.

8. The studies do not provide insights into how TLR8 is inducing HIV. Is this a direct effect on HIV-1 transcription and infection? Does it reflect a feedback mechanism, such as induction of IL-6 or IFN γ ?

Point-by-point response to reviewers' comments:

We would like to thank the reviewers for carefully reading and commenting on our manuscript, and also for constructive critics and valid questions regarding our results and conclusions. We have addressed the criticism by providing additional controls, statistical analyses and new data to substantiate our claims. The text is changed accordingly and re-written to better explain the relevance of our findings for HIV infection and, in a broader sense, for T cell immunity.

Despite modest effects of TLR8 intrinsic activation of T cells we believe our results are relevant for HIV infection. High concentrations of T cells reside in secondary lymphoid organs where viral dissemination occurs. T cell intrinsic TLR8-activation can thus contribute, directly and indirectly via cytokine-mediated autocrine/paracrine signalling, to increased inflammation and viral replication leading to dissemination of HIV in untreated patients, or viral “blips” in patients on ART. Further relevance for treatment is suggested in a recent paper where SHIV viral rebound was delayed/prevented in monkeys treated with a TLR7 agonist to activate T cells /reactivate latent SHIV in combination with neutralising Abs (Borducchi Nature2018, PMID 30283138). Our results suggest TLR8 agonists should be explored in similar “shock-and-kill” approaches.

We have addressed the critics as follows, reviewers' comments in italics:

Reviewer #1 (Remarks to the Author):

very nice paper and interesting molecular aspects. I am convinced that the data in this paper add substantially to mechanistic insights in TLR8 signalling in CD4+T-cells and potential effects on HIV replication.

However, the data should be positioned in a bigger perspective. The lymphoid system is more than CD4+ T-cells alone, and the effects of other innate immune cells which are triggered by TLRs predominate or even overshadow completely the effects shown here.

We greatly appreciate reviewer's positive assessment of our work. We agree with the reviewer that other immune cells expressing TLR8 will respond to endosomal ssRNA/HIV with cytokine secretion, and in particular myeloid cells like monocytes/macrophages and DCs. Whereas plasmacytoid DCs sense endosomal HIV-1 ssRNA by TLR7 and induce protective type I interferon responses (O'Brien, PMID: 21339641; Luban, PMID: 23084911), myeloid DCs and macrophages sense endosomal HIV-1 ssRNA by TLR8 but fail to induce a protective immune response (e.g. type I IFNs) (Khatamzas, PMID: 28923824; Gringhuis, PMID 20364151; Guo, PMID: 24939850). We also did not observe type I IFN induced by TLR8 engagement in CD4+ T cells (data not shown).

The final outcome will depend on the organ and stage of infection since the cellular composition and activation status will vary, thus the in vivo situation is hard to predict. HIV dissemination occurs in lymphoid tissues where high concentration of T cells are presents. DCs are thought to seed HIV infection in the lymph nodes by transmitting the virus to T cells during antigen presentation. However, productively infected T cells are highly migratory and can disseminate the infection through cell-to-cell transmission locally in the lymph node. In addition, recirculation of HIV-infected T cells is important for the establishment of systemic infection (Murooka, PMID: 22854780). In this situation, the local cytokine environment may

well be shaped by T cells and act in an autocrine/paracrine manner. We added 0.3-3% T cell-depleted PBMCs or purified monocytes to T cells before TLR8 activation. Although this increased IL-6 secretion, the T cell contribution was still substantial and IFN γ production was not affected by the myeloid cells. (Supplementary Fig. 2d & e, where S2d is new in the revised manuscript and shows the contribution to IL-6 production).

Secretion of cytokines other than IL-6 was low (compared to what you would get from myeloid cells), but, as stated in the manuscript discussion page 36, IL-6 has a central role in HIV infection. IL-6 promotes Th17 and inhibits Treg differentiation, and acts as a co-stimulatory factor enhancing proliferation and cell survival by counteracting activation-induced cell death (reviewed in Dienz, PMID: 18845487). A recent study shows that IL-6 is a stronger predictor of fatal events than C-reactive protein during HIV infection (Borges, PMID: 27132283). As recommended by the reviewer and the Editor we have included a paragraph in the discussion page 36, 41 and 42 in the revised manuscript to position our findings in a bigger perspective.

Minor comments:

1) an additional control for the endosomal uptake of HIV would be the blocking of the endosomal uptake machinery and thereupon the lack of HIV in the endosomes.

We agree with the reviewer and to address this, we pre-incubated T cells with Dynasore before co-culture with HIV-iGFP infected HEK293 cells to allow for synaptic transfer in the presence of co-receptor inhibitor to prevent fusion. HIV uptake was inhibited by Dynasore (2.5-fold decrease in frequency of GFP+ T cells). These results are included as Fig. 1d in the revised paper.

2) please, elaborate on the differences of the TLR8 agonists used as they show substantial differences in their effects

The natural ligands for TLR7 and 8 are ssRNAs in the forms of degradation products, nucleosides (guanosine and uridine, respectively), and oligoribonucleotides (Tanji et al., PMID: 23520111 & PMID: 25599397; Zhang, PMID: 27742543 & PMID: 30566863) engaging different binding pockets. CL75, CL264, R837 are synthetic imidazoquinoline derivatives (IQDs) that are able to strongly interact with TLR7 and 8 homodimers and activate signalling in the absence of oligonucleotides, which are believed to function mainly to enhance the binding affinities of the nucleosides/IQDs. Poly U is able to activate TLR8 alone, but interactions are stronger if combined with degradation products (uridine, endogenous nucleoside derivatives or IQDs). Thus, although differences in potency was expected, we were equally surprised by the consistent differences observed between CL75 and poly U, where poly U was most efficient in HIV reactivation and in upregulation of T cell activation markers (Fig. 3b and 7a) whereas CL75 was most potent in activating HIV replication (Fig. 7c). Batch variation could possibly cause some of the differences in potency since experiments were done over a few years. However, our data suggest that IQDs and pU differ in how efficient different signalling pathways are induced.

Other studies have also pointed to differences in signalling between different TLR7/8 ligands (Gorden, PMID: 15661881). Human PBMCs respond to R848 and ssRNA with proinflammatory cytokines but only ssRNA was able to induce type I IFNs (Heil, PMID: 14976262) and recent studies suggest ligand-induced trafficking of TLR containing

endosomes dictates different responses (Miyake, PMID: 29452403; Saitoh, PMID: 29150602; Sasai, PMID: 20847273). The ligands also have different solubility; poly U is membrane impermeable and needs to enter the cell by endocytosis whereas IQDs are lipid soluble and could engage TLR8 in compartments different from poly U (TLR7/8 are processed into active dimers by convertases that are active at neutral pH). Both poly U and CL75 are considered more specific towards TLR8 than TLR7, although structural binding studies show that some cross-specificity is expected (Zhang, PMID: 30566863) which is also what we observed in our study (Figs. 2c, d). This could also affect the overall response. However, differential signalling requirements is best studied for TLR7 and TLR9 in DCs and needs to be explored for TLR8 and in other cell types. In our study, responses from pU/pLA would probably best reflect what happens during HIV infection. A paragraph is added to the discussion in the revised manuscript page 36 and 37.

3) please, provide statistics to fig 3b

This is now done by calculating the area under the curve (AUC) followed by one-way ANOVA with Dunnett's post-test for multiple comparisons).

Reviewer #2 (Remarks to the Author):

Stimulation of TLR8 in antigen presenting cells enhances Th1 and Tfh cell responses, but several studies have also described T cell-intrinsic functions of TLR8.

In this study, Meås et al. report that TLR8 activation in purified human CD4 T cells leads to enhanced T cell activation and pro-inflammatory cytokine production. They propose that endosomal uptake of HIV-1 by cell-to-cell transmission activates TLR8 in CD4 T cells, and they show that stimulation of latently infected CD4 T cells with TLR8 agonists can reverse latency in vitro. While the study addresses an interesting aspect of host-virus interaction, there are several major concerns.

The presented data do not sufficiently prove the claimed TLR8 stimulation by endocytosed HIV-1 in CD4 T cells. Along these lines, the statistical analysis of most of the data in the manuscript is insufficient, which severely compromises the conclusions.

We appreciate the reviewer's acknowledgment of the potential interest of our work, as well as the constructive comments for improvement.

Direct functions of TLR8 in naïve and memory CD4 T cells have been previously reported (Caron 2005, PMID:16034093), particularly in Tregs (Peng 2005, PMID:6123302, Li 2018, PMID:30344014). Here, the authors investigated the effects of TLR8 ligation in primary naïve CD4 T cells. Reversal of HIV-latency by TLR8 stimulation has also been previously demonstrated, albeit not by direct activation of TLR8 in T cells (Schlaepfer 2011, PMID:21357269).

As pointed out by the reviewer T cell intrinsic functions of TLR8 engagement have been studied before with different outcomes and we already referred to these studies in the results page 10 ref. 31, 32 and discussion page 33. The effect of TLR8 ligands on HIV latency reversal was also addressed in the discussion page 32 where reference 62 is the paper by Schlaepfer et al. (PMID: 21357269) and refs. 61+63 are papers from the same group. However, as also pointed out by the reviewer, ours is the first study to show reversal of HIV latency by direct activation of TLR8 in human primary T cells. We were not aware of the recent study by Li et al. (PMID: 30344014) demonstrating TLR8-mediated reversal of Treg

suppressive functions through inhibition of glucose uptake and glycolysis. This is now included in the revised paper as ref. 92.

Specific comments

1. The text claims that Fig. 1 and Fig S1 “confirm that HIV-1 is endocytosed (...) and routed to lysosomes for degradation”. However, Fig. 1B depicts a single LAMP1+/gag-GFP+ punctum. Adequate quantification of LAMP1+ HIV-1 containing endosomes is required to substantiate the claim.

We agree with the reviewer that quantification of LAMP1+ HIV endosomes should have been done. However, we want to point out that we did not base the conclusion on degradation from LAMP1 staining but from Flowsight experiments showing that HIV-iGFP in trypsin-resistant endosomal compartments of T cells decrease over time (Fig. 1c). We attempted to use BafA1 to prevent acidification and thus degradation of HIV, but BafA1 inhibited HIV uptake by T cells as also shown by Sloan et al. (PMID: 23678185).

In a series of new experiments, we have studied endocytosis of HIV (HIV-1 X4-Gag-iGFP, R5-Gag-iGFP or X4-gp120-eGFP) in CD4+ T cells donated by infected HEK293 cells, T cells or macrophages in the presence of inhibitors of CXCR4 (AMD100) or CCR5 (Maraviroc) and of endocytosis (Dynasore). At different time points cells were trypsinized to remove surface bound HIV, stained for HIV-p24, EEA1 or LAMP1 and analysed by confocal microscopy or Flowsight. New results are included in revised Figure 1 and Supplementary Fig. 1. Quantification revealed that acceptor CD4+ T cells contained between 5 and 100 trypsin resistant virus puncta (Fig. 1b) and Dynasore blocked endocytosis of HIV by T cells (Fig. 1d). However, EEA1 and LAMP1 only occasionally stained the HIV+ trypsin-resistant compartments, which came as a surprise. Co-localization events were found in approximately 1 of 20 cells. Some areas/patches close to the plasma membrane stained positive for EEA1 and p24 at early time points, but these were hard to quantify as we could not discern discrete puncta. An example picture is shown in Supplementary Fig. 1a and resembles findings by Bosch et al. (PMID: 18602423) who showed clathrin-dynamin mediated endocytosis of HIV which co-localized with EEA1, but less with LAMP1. Few other studies have assessed EEA1 and/or LAMP1 staining in HIV-infected primary CD4+ T cells. Wang et al. (PMID: 27847357) used FIB-SEM to study virus-containing compartments and found that roughly 4% of HIV puncta were LAMP1+ in the absence of co-receptor inhibitor vs. 7% with co-receptor inhibitor. Our results are in line with these studies.

Taken together we have to conclude that upon co-receptor mismatch, HIV is internalised into a trypsin-resistant compartment that only occasionally stains positive for early (EEA1) or late (LAMP1) endosomal markers. HIV could be transiently passing through EEA1+ endosomes and rapidly degraded in LAMP1+ endolysosomes or, what seems more likely, trafficked through other, less well characterized endosomal compartments in the T cell. If TLR8 signalling originates from this compartment or if the viral RNA is trafficked further, is currently not clear. Neither is the environment requirement for TLR8 activation: endosomal TLR8 dimers are cleaved by neutral proteases, and studies in myeloid DCs have shown that retaining HIV or TLR8 ligands in early endosomes by preventing maturation, increases activation (Khatamzas et al., PMID: 28923824). In fact, critical reading of the literature shows that many studies never address the localisation of endosomal ligands/pathogens and just assume the follow regular endolysosomal trafficking pathways that are best characterized in macrophages and cell lines – and not in T cells. Thus, we apologize for the premature

conclusions drawn in the initial manuscript and want to thank the reviewer for pointing this out so that correct conclusions are made in the revised manuscript. We have rewritten the revised manuscript accordingly Results page 6 and 7-9 ; Discussion 32-34.

2. Most figures lack sufficient statistical analysis or present data from single experiments.

a. Fig. 2A, Fig. 3B, Fig. 4, Fig. 5C lack statistical analysis.

We have repeated experiments with more donors and included statistical analyses as advised by the reviewer for most figures. A few supplementary figures show data from only one or 2 donors as indicated. Statistical analyses used for each figure are stated in the legends and the statistics section in Methods is updated accordingly in the revised manuscript.

Fig 2a has data from 4 individual experiments/donors; Fig 3b (now Fig 3 since Fig 3a is moved to Fig 2d as recommended by reviewer 3), Fig 4 and Fig 5c show data from 3 individual experiments/donors. Statistical analyses are included (AUC followed by one-way ANOVA with Dunnett's post-test).

b. Fig. 2A and Fig. S2C-E show individual values without SD or indication of replicates. Are these single replicates (n=1)?

Fig 2a now shows data from 4 individual experiments with statistics (AUC followed by one-way ANOVA with Dunnett's post-test). Fig S2c shows data from one donor. The purpose of the figure is to confirm the findings in Fig 2a (which is done with 4 donors) with CD4+ T cells purified by different methods. Fig S2c was done twice with similar qualitative results, but large differences in the magnitude of the IL-6 response between the donors would yield huge error bars, so we would like to show only one donor. Fig S2d (Fig S4a in the revised manuscript) shows data from one experiment/donor. This is the only experiment where we could detect (low amounts) of IL-6 after siRNA treatment of CD4+ T cells. The siRNA treatment was toxic to the T cells and surviving cells were less responsive to activation. We have emphasized this in the revised manuscript together with discussion of the CRISPR/Cas9 knockdown of TLR8, page 11 and 35. We still think it is interesting for readers to see these data. Fig. S2e (as well as new Fig. S2d) were repeated with more donors and now show combined data from three (Fig. S2e) or five (new Fig. S2d) individual experiments with statistical analysis (one-way ANOVA with Dunnett's post-test).

c. Fig. S5A shows "one representative experiment repeated three times with similar results", no statistical analysis.

Fig. 5c and the associated Fig. S5a (now Supplementary Fig. 6c) now show combined data from 3 (Fig. 5c) and 5 (Supplementary Fig. S6c) individual experiments/donors with statistical analysis (AUC for 5c, one-way ANOVA with Dunnett's post-test for S6c).

d. Data in Fig. 2D, Fig 3A, Fig. 5D, Fig. 6 (Fig. 7?) were analyzed by student's t-tests. However, t-tests are inappropriate when comparing more than two experimental groups.

We agree with the reviewer and have redone all statistical analyses using one-way ANOVA with Dunnett's post-test (Fig. 2c; Fig. 3a which is now Fig. 2d; Fig. 6 and Fig. 7) and two-way ANOVA with Dunnett's post-test (Fig. 5d).

e.Fig. S7 shows one experiment performed on pooled cells from four aviremic patients. It is unclear how many of the donors responded to the treatment.

Findings in Fig. S7 were repeated with T cells from 5 more aviremic HIV patients. In addition, latency reversal agents SAHA and Bryostatins were included in these experiments as recommended by reviewer 3. The new data with statistics are incorporated in Fig. 7a of the revised manuscript (one-way ANOVA with Dunnett's post-test), and Fig. S7 is removed.

3.TLR8-specific inhibitors were used to block HIV1-induced cytokine production (Fig. 6 /S6). While both inhibitors blocked CL075- and pU/pLA-induced cytokine production (Fig. 2D), only CU-CPT9b inhibited HIV-1-induced IL-6 production (and only when comparing log2 fold change, not when comparing absolute values, Fig. S6). There seems to be little effect of TLR8 inhibition on HIV-induced IFN γ production, despite IFN γ being induced by TLR8 agonists and inhibited by TLR8 inhibitors (Fig. 3). IL-1 β production is potentially the most striking effect of TLR8 stimulation in CD4 T cells (Fig. 3A), yet, IL-1 β response to HIV-1 infection was not assessed.

We recently developed a series of highly potent and specific small-molecule TLR8 inhibitors. First generation compounds CU-CPT8 – 9a and the control compound 6 are published in Nature Chemical Biology (PMID: 29155428), and second-generation compounds CU-CPT9b-f are published in Cell Chemical Biology (PMID: 30100350). Although all compounds are specific for TLR8 and of similar potency in some assays, we consistently obtained better inhibition of TLR8 activation in T cells with CU-CPT9b than 9a in our experiments, including HIV-induced responses.

Due to donor variations results in Fig. 6 are normalized to their respective uninfected controls without or with compounds, with absolute values in Fig. S6. Data were only plotted with a log2 y-axis, this is now changed to linear. We have performed 3 new experiments also including a higher concentration (25 μ M) of the compounds and statistics are re-done as recommended by the reviewer using one-way ANOVA with Dunnett's post-test. Absolute values with statistics are now in Supplementary Fig. 7, and we do get significant reduction of IL-6 and IL-17 with 5 μ M CU-CPT9a and b from X4 or R5 tropic HIV since more experiments are included. Since only 3 donors are included in experiments with 25 μ M inhibitor, significant reduction was only obtained when data were normalized to their respective controls (Fig. 6). We suggest including these results in the revised manuscript and take out IL-10 results (we can include IL-10 if the reviewers/Editor prefer it, results have not changed, but the paper in general focuses more on IL-6, IFN γ and IL-17).

The reviewer correctly states that inhibition of synthetic TLR8 ligands was more complete than with HIV. However, we are not claiming that TLR8 is the only receptor on T cells responding to HIV. We only claim that when HIV is endocytosed and cannot fuse into the cell cytosol, it will engage TLR8 in CD4+ T cells. HIV-1 can activate additional PRRs that may contribute to the results we show, although few studies are done on T cell intrinsic activation (reviewed in Iwasaki, PMID: 22999945; Silvin, PMID: 25617674). In productively infected cells HIV could activate cytosolic RNA/DNA sensors, but with efficient blockade of co-receptors this should not happen. We also don't observe a type I IFN response that you would expect from such interactions.

A series of studies by Warner Greene's group have shown that DNA products generated in abortively infected CD4+ T cells activate inflammasomes resulting in IL-1 β secretion and pyroptosis, whereas Claudia Kemper has shown that NLRP3 inflammasome activation in human CD4+ T cells is dependent on intracellular complement and contributes to IL-1 β and IFN γ secretion (Arbore, PMID: 27313051). We are also enthusiastic about the IL-1 β secretion induced by TLR8 ligands since this could imply inflammasome activation resulting from TLR8 engagement, as has been suggested by others in monocytes (Guo, PMID: 24939850; Chattergoon, PMID: 24788318). We did assess IL-1 β secretion from HIV infected cells in the experiments shown in Fig. 6, but unfortunately results were negative and thus not shown. From the ligand studies we observed very high donor variations (1 – 800 pg/ml IL-1 β), but TCR activation alone did not induce IL-1 β and the response was efficiently taken down with TLR8 inhibitors (Fig. 3a/new Fig. 2d and Supplementary Fig. 5). Our results suggest CL75 was more efficient than poly U, which would be closer to viral RNA, but we currently don't know why HIV does not induce IL-1 β . One possibility could be that HIV actively represses inflammasome activation in T cells, but this will have to be explored in future studies. We have discussed differences between TLR8 ligands and ligands vs HIV in the revised manuscript, page 36 and 37.

The proposed TLR8 stimulating effect of HIV-1 is not convincingly proven by the data in their current state. Genetic approaches may yield more definitive results, and several methods for gene editing of primary T cells have recently been published.

It is well known that gene editing of primary T cells is difficult to achieve. In a series of new experiments with varying conditions we have knocked-down TLR8 in resting and TCR-activated CD4+ T cells by CRISPR/Cas9 ribonucleoprotein electroporation. Genetic modification of TLR8 was detected using a Genomic Cleavage Detection Kit. Cells were rested 5-16 days post-electroporation to allow for protein turnover and to ensure that they entered into contraction phase prior to re-stimulation, then re-stimulated with aCD3/aCD28 bound plates or TCR activation-beads in the presence of CL75, pU/pLA or FSL-1. However, all samples (including cells that did not undergo electroporation) failed to respond adequately to TCR re-stimulation and TLR8 ligands. Prolonged resting time improved responsivity but cells still produced 10-50-fold lower IL-6 and IFN γ and massive cell death was observed after re-activation in all cases (70-90%). Thus, despite successful knockdown, our results suggest that this lengthy protocol, requiring two times TCR activation, results in toxicity upon re-activation (possibly due to activation-induced cell death (AICD)) and non-responsiveness in the surviving cells. Example figures below show results from 10-20 % surviving T cells.

In a very recent preprint article in bioRxiv (Aksoy 2019 doi: 10.1101/466243), it was shown that CRISPR/Cas9 gene editing can be done in resting T cells using higher voltage electroporation than used for pre-activated cells due to their smaller size. We thus electroporated resting cells from 2 donors using these parameters and obtained about 50% knockdown efficiency for donor 1 and 37% for donor 2 (day 5, calculated from Genomic Cleavage Detection Kit). Cells were TCR-activated in the presence or absence of TLR ligands with 60-80% of the cells surviving. TLR8 KD reduced IL-6 and IFN γ induced by CL75 and pU/pLA, but not FSL-1, compared to control gRenilla. However, the treatment still dramatically affected cell responsiveness (see figures below). In comparison to untreated cells, cells electroporated with control gRenilla showed 2-5-fold reduction in IL-6 and IFN γ secretion even to TCR activation alone. This raises concerns that using CRISPR/Cas9 in CD4+ T cells compromises cell functions, and we did not proceed with HIV infection. However, the results confirm TLR8-specific activation of CD4+ T cells and we propose to include a new Supplementary Fig. 4 in the revised manuscripts where some of these results (a, b, d, g) are presented together with the siRNA data. We chose not to include the untreated cells in the figure since they were not subject to CRISPR/Cas9 electroporation (gRenilla serves as a control) – but we show them all here for your information:

4. It is curious that pU/pLA reactivated viral replication in latently infected CD4 T cells from all donors, while CL075 did so only in one donor (Fig. 7A). In contrast, CL075 enhanced productive HIV infection, which pU/pLA did not (Fig. 7C). How do the authors explain these differences?

We agree with the reviewer, please see the response to reviewer 1, minor point 2

Minor comments

1. Residual content of 0.27% of CD14+ cells in the T cell preparations could potentially account for the observed IL-6 production in response to TLR ligands.

Possible contamination from other cells has been a concern throughout our study, thus we have taken care to only include experiments with T cells of high purity. Still, as pointed out by the reviewer, some residual cells are left. The frequency of CD14+ cells from positive T cell isolation was 0.27% whereas it was even less (0.18%) from negative isolation, which was used for all experiments (Supplementary Fig. 2b). We have done new experiments addressing the possible contribution to IL-6 secretion by adding up to 1% PBMCs or CD14+ monocytes to the T cells during TCR+TLR ligand activation. These results are included in Supplementary Fig. 2d and show that although monocytes – as expected – respond to TLR8 ligands with IL-6 production, the CD4+ T cell contribution is still significant.

2. Fig. 2C shows the knock-down efficiency of siRNA transfected T cells. However, the only panel analyzing TLR8-silenced cells is in Suppl. Fig. 2D.

We have now moved all siRNA data to the new Supplementary Fig. 4 together with the CRISPR/Cas9 gene editing data. The western blot showing that TLR8 protein is present in CD4+ T cells is kept in the main Fig. 2b.

Reviewer #3 (Remarks to the Author):

Meas et al examines the expression and function of TLR8 in primary CD4 T cells. Furthermore, they explored the potential that HIV-1, through an endosomal route of entry can activate signaling through TLR8. Experiments include manipulating receptor function using a panel of TLR agonist and antagonist and assessing T cell function and HIV expression and replication. Major findings include showing that TLR8 is expressed in CD4+ T cells, that TLR ligands elicit T cell responses including the induction of cytokines and phenotypic markers that may reflect maturation of cells into TH1 and TH17 cells and that TLR8 can facilitate active HIV-1 replication and latency reversal. Strengths of the paper includes the exhaustive characterization of responses and the use of a library of TLR ligands and inhibitors to map out the role of TLR8. However, many of the TLR mediated responses are modest, the responses to ligands and inhibitors are highly variable, mechanisms of action are not explored and there are some concerns about the relevance of trans-infection models that are being used. Overall, the data are intriguing, they raise a number of relevant questions but the study is preliminary and requires controls and additional mechanistic studies. Some specific comments are below.

We greatly appreciate reviewer's acknowledgment of the importance and potential interest of our work.

1. Fig 1 shows transfer of HIV to a target cell by an endosomal route. For this experiment HeLa cells are used to transfer HIV which raise some concerns about the relevance of the cell type. In addition, the results are not well described and appears that the process of entry is inefficient. For example, in fig 1B only one particle of four appears to be associated with endosome/lysosome. The number of particles associated with a synapse or a LAMP-1 compartment should be quantified since they are showing only selected images of a single or few cells. A minor point is that it seems these data should be presented with the other experiments that address HIV endosomal entry and TLR8 signaling (Fig 6 and 7).

We agree with the reviewer's concerns and refer to our response to reviewer 2's first comment regarding EEA1/LAMP1. New experiments and more detailed analyses revealed only occasional EEA1/LAMP1 staining of HIV endosomes and made us change our conclusion accordingly. These experiments were done with more relevant donor-cells as well, and in the revised Fig. 1a we have new pictures showing virological synapses formed between human monocyte-derived macrophages (MDMs) and CD4+ T cells and between infected and recipient CD4+ T cells, in addition to the picture from the original Fig. 1 of HEK293 cells and CD4+ T cells. The synapses form frequently and involve several HIV puncta or even patches of virus, but these are hard to quantify. However, in new Fig. 1b we show the HIV puncta (p24 staining) in trypsin-resistant compartments of recipient CD4+ T cells subsequent to trans-infection, quantified to 5-100 HIV puncta per cell (Fig. 1b). Flowsight data from the original Fig. 1 (graph in Fig. 1c, pictures moved to Supplementary Fig. 1b) similarly showed 20-30% HIV-iGFP positive T cells (trypsin treated after trans-infection) that declined with time (chase), suggesting HIV degradation. Co-receptor inhibitors AMD3100 or Maraviroc were included in all experiments to prevent fusion. We also included Dynasore to prevent endocytosis and show in new Fig. 1d that this decreased HIV-iGFP in recipient CD4+ T cells. Together the new data show efficient endocytosis of HIV in CD4+ T cells donated by infected HEK293 cells, MDMs or other CD4+ T cells.

Figure 1 was originally done to confirm studies by others showing trans-infection and endocytosis of HIV in T cells since this is a prerequisite for our hypothesis that HIV can interact with TLR8 in T cells, in particular when there is co-receptor mismatch. We have discussed the sequence of figures back and forth but prefer to keep them in the current sequence, opening and ending with HIV, unless reviewers/Editor agree on moving them.

2. Much of the paper focuses on TLR8 signaling and the potential that this influences CD4+ T cell function. Figure 2 addresses TLR8 expression with western blots and induction of IL-6. Actually, the IL-6 data should be presented in figures that includes induction of the other cytokines since that addresses function rather than expression.

We agree with the reviewer and have moved cytokine data from Fig. 3a to new Fig. 2d.

It would have been interesting to look at T cell subsets, such as memory cells, especially because of the importance of these cells in HIV latency.

We thank the reviewer for pointing this out, and we have now analysed IFN γ -production in response to TCR-TLR stimulation from CD4+ T cell subsets with a memory (CD45RO+CD45RA-) and naïve (CD45RO-CD45RA+) phenotype (flow-cytometry at 72h, 6 donors). The results have been added to the revised manuscript as Supplementary Fig. 6a. In

accordance with other studies (Caron, PMID:16034093) we find that most cells that produce IFN γ are memory CD4 $^+$ T cells.

It was not clear as to the importance of the siRNA knock-downs since these cells and knockdowns were not used in the paper.

The siRNA knock-down was mainly included as a control for the detection of TLR8 in CD4 $^+$ T cells. However, cells were also used for initial experiments to inhibit TLR8-induced responses as shown in Fig. S2d (now Supplementary Fig. 4), but these experiments were discontinued due to toxicity of the treatment. This is discussed in the revised manuscript together with discussion of the CRISPR/Cas9 knockdown of TLR8 in page 11 and 35.

3. The cytokine expression data in Fig 2 and 3 should be combined.

Cytokine data are now moved from Fig. 3a to new Fig. 2d.

Other than IL-1 β and IL-6 and maybe IFN γ , the induction of cytokines and markers are very modest, usually less than 2 fold. It would be helpful in appreciating the robustness of this response to include positive controls such as CD3+CD28, PHA or PMA.

CD3/CD28 (TCR activation) was included in almost all experiments as we saw the greatest effect of TLR8-stimulation in TCR-activated cells (Fig. 2a). Thus, although TLR8 ligands by themselves could induce some responses in resting cells, we mostly address the adjuvant contribution of TLR8 engagement which ranged from nothing (IL-2, 4, TNF) to 2-50 fold with IL-6 and IL-1 β being the highest (Fig. S5, S6). TCR activation by itself efficiently induces IFN γ and TNF and the adjuvant effect of TLR8 activation was modest, as pointed out by the reviewer. We have tried to make this clearer in the revised manuscript, page 34-35.

Of interest, a paper was just published in Science Signaling by Rodríguez-Jorge et al. (PMID: 30992399) used modelling to show cooperation between TCR and TLR5 signalling for CD4 $^+$ T cell activation. Although we study TLR8 responses and HIV is not expected to engage TLR5, most of the TLR signalling pathways are similar and thus their findings regarding cross-reactivity of the TCR and TLR5 signalling pathways are probably valid here as well.

We often included PMA/ionomycin in the flow cytometry experiments assessing intracellular cytokines where PMA/ionomycin induced stronger cytokine responses than TCR+TLR8. However, we believe that aCD3/aCD28 TCR activation is more physiological relevant than PMA/ionomycin. Optimal cytokine secretion as well as intracellular cytokine production was induced 24-72h post aCD3/aCD28 TCR activation – with and without TLR ligands – whereas PMA/ionomycin was only added for 4-6h due to toxicity. An example is shown here, but we will not include this in the revised manuscript unless the reviewer/Editor ask for it:

4. Fig 4 the authors conclude that TLR8 acts in **synergy** with TCR. However, the cellular responses are modest and there are not statistics to support the conclusion of synergy. How is synergy being defined?

We agree with the reviewer that we don't show synergy but rather that TLR8-activation increases some of the TCR responses. We have corrected the manuscript accordingly to say "increased", "enhanced" etc. for the adjuvant TLR8 activation.

5. An interesting conclusion of the paper is that TLR8 signaling maybe biasing cell function towards TH1 cells and possibly TH17 cells, although the latter data are less robust. Expanding this observation and the mechanism driving this maturation would strengthen the paper.

We appreciate that the reviewer finds the results interesting. TLR8 stimulation increased TCR-activated production of Th1 and Th17 lineage effector cytokines (IFN γ and IL-17) with maximum responses at 48h (Figs. 5c, d) which could be significantly inhibited by TLR8 inhibitors (Fig. 5d). These experiments do not formally reflect a bias towards Th1/Th17 maturation/differentiation, but rather indicate an increased ability of (mainly memory) T cells to produce Th1/Th17 lineage-specific cytokines in the presence of TLR8 stimulation (while production of other T cell cytokines such as TNF, IL-2 or the Th2 lineage cytokine IL-4 were not increased). To more directly address the effect of TLR8 stimulation on differentiation/maturation towards Th1/Th17, we performed in vitro expansion and maturation experiments with re-stimulation on day 7 and obtained similar results (Supplementary Fig. 5a, now 5c in the revised manuscript). More important, TLR8 stimulation increased the expression of the lineage-specific transcription factors T-bet (Th1) and ROR γ T (Th17) in IFN γ and IL-17 producing T cells, respectively, which was significantly reduced with TLR8-specific inhibitors (Supplementary Fig. 5b). Mechanistically this is most likely driven by TLR8-induced IFN γ and IL-6 in a paracrine/autocrine feedback loop: IFN γ is known to induce T-bet (STAT1 signalling) and IL-6 induces ROR γ T (STAT3 signalling). Interestingly, IL-6 in this scenario is often said to originate from innate cells whereas we show that T cell intrinsic TLR8 activation can induce this feedback loop. These points are now included in the discussion of the revised manuscript, page 35. We cannot exclude, however, that TLR8 signalling has direct effects on Th1/Th17 differentiation.

6. The experiments examining whether endosomal entry of HIV trigger TLR8 signaling (Fig 6) are modest with the exception of IL-6.

We agree with the reviewer and, as pointed out in our response to comment 3, TCR activation by itself efficiently induces IFN γ and the adjuvant effect of TLR8 activation was modest. This was most evident in HIV infection experiments where, dependent on how strongly the donor responded to TCR activation, IFN γ levels were very high before cells were infected and the contribution from the virus was modest (Fig. 6 and S7). However, both X4 and R5 HIV significantly increased IL-6 and IL-17 production in a TLR8 dependent manner, suggesting TLR8 signalling pathways more strongly induce these cytokines. We have tried to make this clearer in the revised manuscript in the results, page x, and in the discussion, page 34-35.

In addition, and as pointed out in the response to reviewer 1, even modest cytokine levels can contribute significantly to autocrine/paracrine activation e.g. in secondary lymphoid organs where T cell proliferation /numbers are high, and HIV can disseminate through cell-cell interactions. IL-6 has a central role in HIV infection. It promotes Th17 and inhibits Treg differentiation, and acts as a co-stimulatory factor enhancing proliferation and cell survival by counteracting activation-induced cell death (reviewed in Dienz, PMID: 18845487). Furthermore, IL-6 has been reported to promote NFATc2 expression and NFATc2 can enhance HIV proviral expression by binding to the HIV LTR (Diehl, PMID: 12093869; Romanchikova, PMID: 14748509).

Furthermore, the variation in the activity of the different inhibitors was not discussed.

Please see our response to reviewer 2, point 3.

Proviral levels should be measured to confirm that infection is not being established. Like HeLa cells, there are concerns about using HEK293 cells and it would be informative to confirm these results with macrophages or dendritic cells.

We agree with the reviewer that more relevant HIV infected donor cells should be used for trans-infection and these results are now included in revised Fig. 1 and Supplementary Fig. 1. The co-receptor inhibitors AMD3100 and Maraviroc were chosen since they have proved efficient in several studies of HIV transmission in T cells at concentrations used in our study as well (Sloan, PMID: 23678185; Blanco, PMID: 15371410; Bosch, PMID: 18602423). Proviral levels can be detected by PCR to confirm the absence of HIV integration. However, in cell-to-cell transmission experiments, the donor cell is productively infected with HIV and even few numbers would contaminate the provirus PCR analysis of recipient CD4+ T cells. Therefore, it would be essential to re-isolated highly pure acceptor CD4+ T cells from the co-culture, which is very difficult to do in this infection model due to the presence of donor/acceptor virological synapses. We never saw any evidence for productive infection of recipient T cells using the co-receptor inhibitors throughout our imaging and Flowsight experiments, and since we use HIV-GFP we could compare to how infected cells looked.

7. Experiments in Fig 7 suggest that TLR8 may influence HIV expression and latency reversal. A minor point is that technically they are measuring induction of RNA levels and not the outgrowth of virus, which is detected by expanding induced virus by infecting indicator CD4+ cells.

We performed viral growth assay to test the effect of TLR8 ligands on latency reversal. In our experiment, CD4+ lymphoblasts were used to expand induced virus. P24 ELISA has been used to estimate virus outgrowth in the supernatant (Laird, PMID:23737751). Due to low sensitivity of ELISA, we assessed virus propagation by measuring cell-free virus copy numbers using a qPCR-based method. In fact, this was done in the hospital along with clinical samples for HIV diagnostics. However, indicator cell lines can also be used for that purpose.

It was somewhat surprising that PHA without TLR8 ligands did not induce HIV RNA from the ART samples. It would be informative to include other known latency reversing agents to get a sense of how efficient TLR8 ligands are at inducing HIV transcription and whether TLR8 agonist could synergize with these compounds. Proviral load should be measured since increase replication might reflect greater HIV infection

We agree with the reviewer and included 5 more HIV patients for a new latency reversal experiment and tested the HDACi SAHA, the PKC agonist Bryostatins and PMA/ionomycin in addition to the TLR ligands. Despite slight increase of viral re-activation in response to PHA, TLR8 /poly U activation is still the most efficient to reverse latency when compared to PHA and tested LRA's. These new results are included in the revised Fig. 7. Measuring the proviral load of patients can give indication of the magnitude of input virus which can have an impact on the quantity of expanded virus. However, irrespective of virus quantity, poly U reversed latency in all patients while other treatments did so only in a few of the patients.

These results reinforce the clinical relevance of our study and suggest TLR8 agonists should be considered for "shock-and-kill" strategies alongside LRAs already in clinical trials. In a recent paper SHIV viral rebound was delayed/prevented in monkeys treated with a TLR7 agonist to activate T cells /reactivate latent SHIV in combination with neutralising Abs (Borducchi, PMID: 30283138).

8. The studies do not provide insights into how TLR8 is inducing HIV. Is this a direct effect on HIV-1 transcription and infection? Does it reflect a feedback mechanism, such as induction of IL-6 or IFN γ ?

This is an important point and we think, based on the results we have with TLR8 ligands and knowledge about TLR signalling and HIV transcription, TLR8 activation contributes to HIV replication directly and through feedback mechanisms, in particular via IL-6. HIV transcription involves several transcription factors like NF κ B, AP-1, NFATs. TLR signals activation of NF κ B and AP-1 (via p38 MAPK) which can directly contribute to viral transcription. In addition, TLR8 activation induces IL-6 secretion (and more) which can promote NFATc2 expression, and NFATc2 can enhance HIV transcription by binding to the HIV LTR (Diehl, PMID: 12093869; Romanchikova, PMID: 14748509). We have included this in the discussion in the revised manuscript, page 38, although it will have to be experimentally confirmed and is something we would like to explore further.

In addition, TLR8 activation most likely influences other stages of the HIV life cycle / HIV host-dependency factors as well – and these are harder to predict since NF κ B/AP-1 is involved in transcriptional regulation of so many genes.

Reviewers' comments:

Reviewer #1 (Remarks to the Author):

The authors have addressed the issues I raised in the first review process. Thus, I am fine with it. Nevertheless, studying the paper a 2nd time, I am so free to raise a number of points to consider which might improve the manuscript.

i) def of elite controllers = no HIV RNA detectable

ii) page 5, line 92: MyD88 has been defined already on page 4

iii) the authors do not discuss in the introduction that CD4+ T cells express TLR8, resp. only indirectly by stating their findings at the end of the introduction. Furthermore, my personal preference is to have a working hypothesis and the specific aims scientists address at the end of the intro instead the summary of the findings (as the findings are presented in the abstract and again at the beginning of the discussion, etc).

iv) experiment with dynasore is highly appreciated and the data are nice consistent with the data presented. Nonetheless, I was amazed not see a more complete "black-white" response, i.e., partial response pointing either to a less than absolute blocking of endocytosis by dynasore or HIV enters T-cells even cell-associated by direct fusion - would be nice if authors discuss this issue.

v) I am convinced that data presented in figure 2d go together with data in figure 3, and thus, I would move data figure 2d to fig 3.

vi) line colours in fig 4 are not optimal, i.e., it is difficult to distinguish the black line form TCR vs the dark blud from TCR+TLR7

vii) line 285-288: while correctly written, it could be more precisely stated that there was only a statistical difference for p38 and not Erk.

viii) line 493 to 508: "something" went wrong here; the same paragraph has been presented already in "line 478 to 493.

ix) line 549 to 551: wording - please, check the wording.

x)line 554: please, define HIV infection: natural or ART treated....

xi) line 571: instead of activating, I would prefer enhancing to limit confusion wit "HIV reactivation"

Reviewer #2 (Remarks to the Author):

Comments to the authors.

Meas et al. have taken most of the reviewers' advice and criticism and they have gone to great lengths to improve their manuscript. They have rectified some initial insufficiencies and added valuable new data, such as additional imaging analyses to strengthen their claims and to underscore their hypothesis. In general, I feel that the findings are of interest. This said, some serious issues still remain, which I have pointed out in the specific comments below.

My main concern is again centred around quantification and (over-)interpretation of data. I am commenting on these issues not out of mistrust towards the authors, but rather because I believe that transparency and reproducibility of the data is in the authors' and the prospective readers' best interest.

Specific comments:

1. Fig. 1c,d and S1c,d still lack statistical analysis, error bars and indications of replicates. Are these single donor experiments? If so, these need to be repeated.

2. In their reply, the authors state that RNAi treatment of T cells was highly toxic, such that they could only detect IL-6 production in a single experiment (out of an unknown number of repeats), which is why they chose to show only one experiment. They also state in their reply that they experienced similar problems (incomplete knockdown and high toxicity and unspecific effects of electroporation) when they employed CRISPR/Cas9 gene editing. The results shown in Fig S4 indicate no statistics or number of replicates.

It is clear that the authors made a serious attempt to address the role of TLR8 in CD4 T cells by loss-of-function approaches. However, these attempts appear to have failed due to technical difficulties. I would therefore avoid conclusions or suggestions, especially when based on single experiments, which may very well be experimental outliers. I therefore strongly suggest to either improve the experimental design (which may be difficult) or to remove these data (RNAi and CRISPR) from the manuscript.

3. With the exception of P-p38, the data in Figure 4 show no statistically significant differences between TCR and TCR+TLR7/8 activated T cells. I understand that this may be largely due to inter-donor variation and a limited number of repeats. However, I think it is important that it is clearly stated in the text, such that the reader can exercise caution when interpreting the results. Statements as “nearly doubled” “increased signalling 20-fold” should be avoided when the data show such a high level of variation.

4. The same applies to Fig. 5. In Fig 5c, the data show no significant difference between the experimental groups for any of the cytokines measured. Yet, the authors claim that “TLR8 stimulation increased production of Th1cytokine IFN- γ and Th17 cytokine IL-17 five- to ten-fold compared to TCR activation alone”. In Fig. 5d, the claimed “five- to ten-fold difference” is not visible. For instance, IFN γ is increased from around 4% to 7%, the same applies to IL-17 (CL75, compare black vs. red bars). A similarly modest effect is shown in Fig. S2e and Fig. S6a and S6c (compare red and black bars), without statistical significant differences.

Thus, these statements are not warranted at best, or even misleading, given the variable results and the fact that no statistical difference was detected. I am well aware of the current discussions around the scientific value of statistical tests and p-values in biomedical research. And I agree to some extent. However, I think it is not scientifically sound to make quantitative statements & conclusions based on data from 3 individual donors that show no statistically significant difference, especially when another set of experiments presented in the same figure shows a much weaker (non-significant) effect. The error bars in Fig. 5c clearly indicate a high level of variability, and the effects could thus well be within the margin of error. These experiments need to be repeated a sufficient number of times and the differences must be accurately determined in order to warrant the claims made in this manuscript.

Minor comments

1. I cannot find the legends to the supplementary figures.
2. Several Figure- and supplementary Figure numbers referred to in the text do not match the updated version of the figures.

Reviewer #3 (Remarks to the Author):

The authors have made significant revisions to the manuscript including adding data, statistics, experimental details and discussion. Most of the data exhaustively focused on describing how TLR7, 8, and 9 signaling influence T cell function and maturation. There are still some concerns about the overall experimental model for HIV-1 infection which is using fusion inhibitors to force HIV into endosomal pathways. How efficient is this pathway especially in the context of receptor mediated infection? If establishing infection through endosomes is a relatively rare event, then this also raises concerns about the potential role of TLR8 mediated inflammation in perpetuating HIV-1 expression, infection and associated diseases. Although this mechanisms of cell-cell transmission is possible, it is just unclear as to whether it actually contributes to HIV-1 infection and disease. However, the potential role as a latency reversal agent of pU/pLA was intriguing and exploring this mechanism would be of interest.

Point-by-point response to reviewers' comments:

We want to thank the reviewers for constructive criticism once more. By addressing their concerns, our conclusions are now better supported by statistically significant data and the manuscript is considerably improved. Several changes are made:

- According to the guidelines, we have changed all figures to show all data points in bar charts where less than 10 donors are included.
- We have included several new donors to repeat experiments where results were not statistically significant.
- We have modified the text according to reviewers' suggestions to clarify uncertainties and to avoid over-interpretation of data, e.g. on the *in vivo* relevance to HIV infection (in humans) which we have not tested.

Changes in text are shown in red in the manuscript file, while changes made in the first revision are kept in blue as before.

Reviewers' comments:

Reviewer #1 (Remarks to the Author):

The authors have addressed the issues I raised in the first review process. Thus, I am fine with it. Nevertheless, studying the paper a 2nd time, I am so free to raise a number of points to consider which might improve the manuscript.

We thank the reviewer for the good feedback on our revision, but also for raising some new points that will improve the final manuscript.

i) def of elite controllers = no HIV RNA detectable

We agree with the reviewer and we have incorporated the definition of elite controllers (page 3 in the revised manuscript).

ii) page 5, line 92: MyD88 has been defined already on page 4

The duplicate definition of MyD88 has been omitted (page 5 in the revised manuscript).

iii) the authors do not discuss in the introduction that CD4+ T cells express TLR8, resp. only indirectly by stating their findings at the end of the introduction. Furthermore, my personal preference is to have a working hypothesis and the specific aims scientists address at the end of the intro instead the summary of the findings (as the findings are presented in the abstract and again at the beginning of the discussion, etc).

We have now included in the introduction a sentence about TLR8 expression in CD4+ T cells with reference to literature (page 5 in the revised manuscript).

Regarding the ending of the intro: since this is suggested by only one of the three reviewers, we suggest changing it only if the Editor wants to and this is the preferred style of Nature Communication papers. An alternative ending could then be: "We hypothesized that HIV lacking the correct cell tropism and thus unable to undergo membrane fusion, is trapped in endosomes of CD4+ T cells. This could result in exposure of the viral genome to endolysosomal TLRs and activation of responses impacting on HIV infection." However, we prefer not to change the ending.

iv) experiment with dynasore is highly appreciated and the data are nice consistent with the data presented. Nonetheless, I was amazed not see a more complete "black-white" response, i.e., partial response pointing either to a less than absolute blocking of endocytosis by dynasore or HIV enters T-cells even cell-associated by direct fusion - would be nice if authors discuss this issue.

Dynasore has been shown to interfere with HIV replication (Song *et al.*, PMID: 31058179), therefore target CD4+ T cells were pretreated with dynasore for one hour and washed prior to co-culture with productively infected HEK293T cell. Since dynasore inhibition is reversible, we expected reduction in endosomal uptake of the virus and not complete inhibition. In addition, dynasore is reported to lose potency in the presence of serum in the cell culture media (Kirchhausen *et al.*, PMID: 18413242). We have previously tested the effect of low- or serum-free media on primary CD4+ T cells. We found that inadequate levels of serum negatively affected cell activation and cytokine responses, therefore the experiment was carried out in complete media which can contribute to partial inhibition of endocytosis by dynasore. We added a paragraph in the discussion (page 17 in the revised manuscript) to clarify this point. The experiment was repeated with 3 new donors (Fig 1 d) and inhibition of HIV endocytosis by dynasore is statistically significant.

v) I am convinced that data presented in figure 2d go together with data in figure 3, and thus, I would move data figure 2d to fig 3.

Originally, data were presented this way but we moved them to Fig. 3 after suggestion from Reviewer 3 in the first revision. We agree with Reviewer 1 and have now moved data from Fig. 2d back to Fig 3a.

vi) line colours in fig 4 are not optimal, i.e., it is difficult to distinguish the black line form TCR vs the dark blue from TCR+TLR7

Colors have been changed to increase the contrast between the different treatments.

vii) line 285-288: while correctly written, it could be more precisely stated that there was only a statistical difference for p38 and not Erk.

We agree with the reviewer and the text has been changed accordingly (page 11 in the revised manuscript)

viii) line 493 to 508: "something" went wrong here; the same paragraph has been presented already in "line 478 to 493.

We apologize and have corrected this (page 17 in the revised manuscript).

ix) line 549 to 551: wording - please, check the wording

In this paragraph we point to different responses induced through TLR7 and TLR8 in different immune cells and include unpublished results that we did not see type I IFNs induced by CD4+ T cells in response to TLR8 ligands. However, if the policy of Nature Communications does not allow us to refer to data not shown, we will remove the sentence on page 19 in the revised manuscript.

x) line 554: please, define HIV infection: natural or ART treated....

We agree with the reviewer that this is unclear, and the text has been changed to "A recent study shows that IL-6 is a stronger predictor of fatal events than C-reactive protein in ART treated patients" (page 19 in the revised manuscript) since the cohort study combined the control arms of the

Strategies for Management of Antiretroviral Therapy (SMART) and the Evaluation of Subcutaneous Proleukin in a Randomized International Trial (ESPRIT) clinical trials.

xi) line 571: instead of activating, I would prefer enhancing to limit confusion with "HIV reactivation"

We agree with the reviewer and the text has been changed accordingly (page 21 in the revised manuscript).

Reviewer #2 (Remarks to the Author):

Comments to the authors.

Meas et al. have taken most of the reviewers' advice and criticism and they have gone to great lengths to improve their manuscript. They have rectified some initial insufficiencies and added valuable new data, such as additional imaging analyses to strengthen their claims and to underscore their hypothesis. In general, I feel that the findings are of interest. This said, some serious issues still remain, which I have pointed out in the specific comments below.

My main concern is again centred around quantification and (over-)interpretation of data. I am commenting on these issues not out of mistrust towards the authors, but rather because I believe that transparency and reproducibility of the data is in the authors' and the prospective readers' best interest.

We thank the reviewer for appreciating our efforts in revising the manuscript. We also agree with the reviewer that transparency and reproducibility is key in science, which in this case is also well taken care of since all raw data are shown in the accompanying Source Data File. Since we work with primary T cells from different donors, we normally see great variation in responses to the various treatments - not only to TLR ligands, but from standard TCR activation as well. Some donors are overall high-responders or low-responders, but with similar trends in response to treatments. Other donors can be high IFN γ -responders but poor IL-17 responders etc. This greatly influences statistics and makes it harder to reach significance for the weaker responses and to all concentrations of treatments and inhibitors. We have included more donors in this second revision in order to meet the reviewer's concerns and to strengthen our claims about the T cell intrinsic effects of TLR8 activation. We have also modified the text throughout the manuscript to better reflect our actual findings without over-interpreting our data and discussed the limitations of our in vitro model in the context of the putative translational implications (i.e. for HIV infection in humans).

Specific comments:

1. Fig. 1c,d and S1c,d still lack statistical analysis, error bars and indications of replicates. Are these single donor experiments? If so, these need to be repeated.

We repeated the experiments in Fig. 1c by flow cytometry with 3 new donors. Experiments underlying Fig 1d were supplied with 3 new donors. Statistical significance was calculated for Fig 1c and 1d. Fig. S1c is a representative dot plot of one of the 3 donors in Fig. 1c. Fig. S1d (previous S1c) shows the frequencies of untreated and TCR-activated CD4 $^+$ T cells expressing CXCR4 or CCR5 and was repeated with 2 new donors. Results for all 3 donors and the mean are shown, but without statistical analysis since this is phenotyping of cells and not really a comparison between treatments or conditions.

2. In their reply, the authors state that RNAi treatment of T cells was highly toxic, such that they could only detect IL-6 production in a single experiment (out of an unknown number of repeats), which is why they chose to show only one experiment. They also state in their reply that they experienced similar problems (incomplete knockdown and high toxicity and unspecific effects of

electroporation) when they employed CRISPR/Cas9 gene editing. The results shown in Fig S4 indicate no statistics or number of replicates.

It is clear that the authors made a serious attempt to address the role of TLR8 in CD4 T cells by loss-of-function approaches. However, these attempts appear to have failed due to technical difficulties. I would therefore avoid conclusions or suggestions, especially when based on single experiments, which may very well be experimental outliers. I therefore strongly suggest to either improve the experimental design (which may be difficult) or to remove these data (RNAi and CRISPR) from the manuscript.

We appreciate that the reviewer understands the difficulties in gene silencing of primary T cells. Despite difficulties we did succeed with gene silencing, but treatments were either toxic (RNAi) or changed the responsiveness (CRISPR/Cas9 gene editing) of the T cells, thus we did not feel comfortable to continue with either protocol. We removed the results (Supplementary Fig. 4 and corresponding text in the methods, results section and discussion) as suggested by the reviewer. A consequence of this is that previous Supplementary Figs. 5-7 are now Supplementary Figs. 4-6 in the revised manuscript.

3. With the exception of P-p38, the data in Figure 4 show no statistically significant differences between TCR and TCR+TLR7/8 activated T cells. I understand that this may be largely due to inter-donor variation and a limited number of repeats. However, I think it is important that it is clearly stated in the text, such that the reader can exercise caution when interpreting the results. Statements as “nearly doubled” “increased signalling 20-fold” should be avoided when the data show such a high level of variation.

We did not mean to over-interpret our results and have changed the text (pages 10 and 11 in the revised manuscript) to more precisely describe the results. We now clearly state that overall phosphorylation kinetics were only for p38 significantly increased by addition of TLR7/8 ligands. However, we still consider it useful for the reader to point at a few further individual observations (increase in NFkB p65 and S6 rp phosphorylation at 5 min stimulation). We describe these findings now more precisely with accurate numbers and state as well that overall kinetics were not significantly changed. Statements such as “nearly doubled” and “increased signalling 20-fold” were removed from the text or replaced with accurate numbers where appropriate.

4. The same applies to Fig. 5. In Fig 5c, the data show no significant difference between the experimental groups for any of the cytokines measured. Yet, the authors claim that “TLR8 stimulation increased production of Th1cytokine IFN- γ and Th17 cytokine IL-17 five- to ten-fold compared to TCR activation alone”. In Fig. 5d, the claimed “five- to ten-fold difference” is not visible. For instance, IFN γ is increased from around 4% to 7%, the same applies to IL-17 (CL75, compare black vs. red bars). A similarly modest effect is shown in Fig. S2e and Fig. S6a and S6c (compare red and black bars), without statistical significant differences.

Thus, these statements are not warranted at best, or even misleading, given the variable results and the fact that no statistical difference was detected. I am well aware of the current discussions around the scientific value of statistical tests and p-values in biomedical research. And I agree to some extent. However, I think it is not scientifically sound to make quantitative statements & conclusions based on data from 3 individual donors that show no statistically significant difference, especially when another set of experiments presented in the same figure shows a much weaker (non-significant) effect. The error bars in Fig. 5c clearly indicate a high level of variability, and the effects could thus well be within the margin of error. These experiments need to be repeated a sufficient number of times and the differences must be accurately determined in order to warrant the claims made in this manuscript.

We agree with the reviewer that the text should precisely reflect the actual data, including if differences were significant (and at what level) or not. We have repeated the experiments underlying data in Fig 5c (4 new donors), Fig 5d (7 new donors), Fig S2e (2 new donors), Fig S6a (new Fig S5b, 4 new donors) and Fig S6c (new Fig. S5d, 7 new donors) and have updated the Figures with statistics. Figure S5 (previous Fig S6) was renamed due to the removal of Fig S4, and the T cell gating strategy was introduced as new Fig S5a (as requested by Nature Communications guidelines).

We can now show that T cell intrinsic IFN γ and IL-17 are significantly increased by TLR8 ligands at 48 h (Fig. 5c and 5d), and responses are significantly reduced by the TLR8 inhibitors (Fig. 5d). Results are updated in a revised Fig. 5.

Fig. S5b (old Fig. S6a) now shows significant TLR8 ligand-mediated increase in IFN γ production in naive as well as memory cells. Fig S5d (previous Fig S6c) shows a significant increase in IFN γ and IL-17 effector cytokine production in CD4 $^{+}$ T cells re-stimulated 8 days after TCR+TLR8 activation (indicating differentiation towards the Th1/Th17 axis).

Fig. S2e shows that TCR+TLR8 ligand treatment significantly increased IFN γ production from T cells. Addition of up to 3% monocytes or PBMCs did not significantly influence the frequencies of IFN γ producing T cells in response to TLR ligands indicating a T cell intrinsic effect of TLR8 stimulation.

Minor comments

1. I cannot find the legends to the supplementary figures.
2. Several Figure- and supplementary Figure numbers referred to in the text do not match the updated version of the figures.

We apologize for the mistakes in not updating correctly the Figure numbers /the text in the revised manuscript, and we have carefully gone over all again to make sure this is corrected in the second revision. We also first forgot to upload supplementary legends and the Nat Com editorial office promised to do it for us as we sent them the file briefly after submission (the site/portal closes once you submit).

Reviewer #3 (Remarks to the Author):

The authors have made significant revisions to the manuscript including adding data, statistics, experimental details and discussion. Most of the data exhaustively focused on describing how TLR7, 8, and 9 signaling influence T cell function and maturation. There are still some concerns about the overall experimental model for HIV-1 infection which is using fusion inhibitors to force HIV into endosomal pathways. How efficient is this pathway especially in the context of receptor mediated infection?

Endosomal uptake of HIV upon the formation of virological synapses between productively infected and uninfected cells has been well documented *in vitro* (Hübner *et al.*, PMID: 19325119, Sloan *et al.*, PMID: 23678185, Miyauchi *et al.*, PMID: 19410541, Ruggiero *et al.*, PMID: 18508887). Endosomal uptake of virus is co-receptor independent and occurs in the presence or absence of co-receptor inhibitors (Hübner *et al.*, PMID: 19325119, Sloan *et al.*, PMID: 23678185). In our study, treatment with co-receptor inhibitors was carried out to prevent fusion from the endosome and thus trapping HIV in endosomal compartments. *In vivo*, a similar outcome can occur when a productively infected donor cell transmits the virus to uninfected cells lacking the appropriate co-receptor. In a humanized mouse model, HIV infected T cells have been shown to be highly migratory and adopt an elongated phenotype in the lymph node where they interact with the surrounding cells (Murooka *et al.*, PMID: 22854780). In addition, normal human tonsils have been shown to contain 66% CCR5- CXCR4+, 6%

CCR5+ CXCR4- and 19% CCR5- CXCR4- lymphocytes (Fig. 3 in Grivel *et al.*, PMID: 17545702). Based on these findings, it is feasible to assume that co-receptor mismatch can occur between cell-associated virus and target cells in the secondary lymphoid tissues. However, to what extent this will occur and influence HIV pathology in patients is difficult to predict and studies are warranted to verify this mechanism in *in vivo* model systems. Sentences/Paragraphs have been added to the discussion pages 16, 21, 24 and 25 of the revised manuscript.

If establishing infection through endosomes is a relatively rare event, then this also raises concerns about the potential role of TLR8 mediated inflammation in perpetuating HIV-1 expression, infection and associated diseases. Although this mechanism of cell-cell transmission is possible, it is just unclear as to whether it actually contributes to HIV-1 infection and disease.

In vitro studies have shown that cell-to-cell transfer of HIV increases the efficiency of virus uptake by an estimate of 18000 fold (Table 1 in Chen *et al.*, PMID: 17728240) and accounts for 60% of total infection based on experimental and mathematical models (Fig. 4. And Iwami *et al.*, PMID: 26441404). A few studies attempted to investigate the contribution of cell-to-cell transmission to HIV infection *in vivo/ex vivo* model systems. Using multiphoton intravital microscopy, Murooka *et al.* examined the formation of virological synapses in the lymph nodes of humanized mice (Murooka *et al.*, PMID: 22854780). The authors showed that productively infected T cells are migratory and adopt an Env-dependent elongated phenotype that facilitate tethering to other cells in the lymph node (Fig 4, suppl. Video 9 and 11). Kolodkin-Gal *et al.* compared the efficiency of cell-free and cell-associated HIV/SIV in mucosal transmission (Kolodkin-Gal *et al.*, PMID: 24109227). In a sealed colon explant model, SIV mac251 infected PBMCs isolated from rhesus monkeys penetrated the colon epithelial layer and established new infection of host cells, while cell-free virus failed to do so (Fig 2). Similar results were obtained in HIV infection of human colonic tissue explants (Fig 4). In addition, cell-associated virus was found to be superior to cell-free virus in initiating infection in rhesus monkeys (Fig 5). Using a humanized mouse model, Law *et al.* showed that cell-to-cell transmission is more efficient in multi-copy infection of target cells with more than one virus genotype (Fig 3 in Law *et al.*, PMID: 27292632). In agreement with Murooka *et al.* (PMID: 22854780), Law *et al.* showed that infected T cells form Env-dependent elongated structures that mediate migratory arrest of uninfected target cells (Fig 5). Taken together, these studies suggest that cell-to-cell transmission of HIV plays an important role in the dissemination of virus in different *in vivo* infection models. However, precisely assessing the contribution of cell-associated and cell-free virus to HIV disease *in vivo* is challenging since both modes of transmission occur concurrently and interdependently. Sentences/Paragraphs have been added to the discussion pages 16, 21, 24 and 25 of the revised manuscript.

However, the potential role as a latency reversal agent of pU/pLA was intriguing and exploring this mechanism would be of interest.

We agree with the reviewer and will follow up with studies on the possible use of TLR8 ligands in latency reversal.

REVIEWERS' COMMENTS:

Reviewer #2 (Remarks to the Author):

The authors have performed addition replication experiments and included statistical analyses where these were missing in the previous version. They have also revised the manuscript to describe their findings more accurately.

The study still contains some discrepancies regarding cellular effects of different TLR8 ligands, endocytosed HIV and TLR8 inhibitors, but these issues have been sufficiently discussed by the authors.

Reviewer #3 (Remarks to the Author):

The authors have addressed most of the comments of all three reviewers. In particular, it is now more clear that the model is in part to drive HIV into an endosomal "dead-end" and triggering TLR8 signaling rather than a path of infection. A minor comment would be to maybe emphasize this rationale in the first sentence or two of the results so that readers focus on the potential signaling and less on whether this is a path of productive infection.

Manuscript NCOMMS-18-34046B

Point-by-point response to reviewers' comments:

We want to thank the reviewers once more for taking their time to review our manuscript. We have addressed the criticism as follows: Changes in text are shown in **green** in the manuscript file, while changes made in the first two revisions are kept in blue and red as before.

REVIEWERS' COMMENTS:

Reviewer #2 (Remarks to the Author):

The authors have performed addition replication experiments and included statistical analyses where these were missing in the previous version. They have also revised the manuscript to describe their findings more accurately.

The study still contains some discrepancies regarding cellular effects of different TLR8 ligands, endocytosed HIV and TLR8 inhibitors, but these issues have been sufficiently discussed by the authors.

We thank the reviewer for the feedback and constructive comments.

Reviewer #3 (Remarks to the Author):

The authors have addressed most of the comments of all three reviewers. In particular, it is now more clear that the model is in part to drive HIV into an endosomal "dead-end" and triggering TLR8 signaling rather than a path of infection. A minor comment would be to maybe emphasize this rationale in the first sentence or two of the results so that readers focus on the potential signaling and less on whether this is a path of productive infection.

We thank the reviewer for the feedback. We have now rewritten the first sentence of the result (p6) to clarify that the infection model was not intended to induce productive infection.